# Olfactory inputs modulate respiration-related rhythmic activity in the prefrontal cortex and freezing behavior

Andrew H. Moberly[1], Mary Schreck[1], Janardhan P. Bhattarai[1], Larry S. Zweifel[2], Wenqin Luo[1] & Minghong Ma [1]

Respiration and airflow through the nasal cavity are known to be correlated with rhythmic neural activity in the central nervous system. Here we show in rodents that during conditioned fear-induced freezing behavior, mice breathe at a steady rate (~4 Hz), which is correlated with a predominant 4-Hz oscillation in the prelimbic prefrontal cortex (plPFC), a structure critical for expression of conditioned fear behaviors. We demonstrate anatomical and functional connections between the olfactory pathway and plPFC via circuit tracing and optogenetics. Disruption of olfactory inputs significantly reduces the 4-Hz oscillation in the plPFC, but leads to prolonged freezing periods. Our results indicate that olfactory inputs can modulate rhythmic activity in plPFC and freezing behavior.

[1] Department of Neuroscience, University of Pennsylvania Perelman School of Medicine, Philadelphia, PA 19104, USA. [2] Department of Pharmacology and Department of Psychiatry and Behavioral Sciences, University of Washington, Seattle, WA 98115, USA. Correspondence and requests for materials should be addressed to A.H.M. (email: amoberly@pennmedicine.upenn.edu) or to M.M. (email: minghong@pennmedicine.upenn.edu)

In addition to autonomic effects[1] and modulation of brain activity via brainstem breathing centers[2], changes in respiration impact neural network oscillations via the olfactory system. With each breathing cycle, nasal airflow entrains the neural activity in the olfactory bulb (OB) and its downstream olfactory cortices[3–5], presumably by activating intrinsically mechanosensitive olfactory sensory neurons (OSNs) in the nasal epithelium[6–9]. When OSNs are functionally compromised or airflow is diverted away from the nose, these respiration-related activities are diminished[10, 11]. Recent studies have led to discoveries of nasal airflow entrained rhythmic neural activity in brain areas beyond the olfactory pathway. The rodent barrel cortex, prefrontal cortex, and hippocampus exhibit oscillations that are phase-locked with breathing and disrupted when peripheral olfactory signals are removed[12–15]. Oscillations in these regions, representing spatially local activity, are also modulated by the respiration-related rhythm. One implication of these results is that aside from its role in physiological homeostasis, the breathing rhythm could also help to coordinate neural activity across multiple brain regions[16].

Low frequency oscillations can serve as a mechanism for long-range communication via synchronization of cortical and subcortical brain regions that are recruited during different behavioral states[17, 18]. This mechanism is utilized during fear discrimination and fear expression when the prefrontal cortex, basolateral amygdala, and hippocampus dynamically interact to mediate appropriate behavior[19, 20]. Specifically, expression of conditioned fear memories is associated with synchronous 4-Hz oscillations in the prefrontal-amygdala circuit that orchestrate neural spiking and predict the onset and offset of freezing periods[21, 22].

Given that the olfactory system is closely linked with limbic brain regions mediating emotion and memory, we investigate the role of respiratory/olfactory rhythms in fear circuitry. Specifically, we focus on the prelimbic prefrontal cortex (plPFC), as a critical region for expression of fear and anxiety[23]. We combine electrophysiology, optogenetics, circuit tracing, pharmacology, and mouse behavior and find that when mice freeze during retrieval of conditioned fear their respiratory rates become steady and center around 4 Hz. This breathing rate is represented in synchronized oscillations in the local field potentials (LFPs) recorded from both the OB and plPFC. Olfactory inputs can reach the plPFC via direct projections from the anterior olfactory nucleus (AON)/ taenia tecta (TT), olfactory structures that receive direct OB inputs, and show respiration-entrained oscillations[24]. Disruption of airflow via unilateral naris closure or ablation of the olfactory epithelium (OE) significantly reduces, but does not completely eliminate the 4-Hz oscillation in the plPFC during freezing. Both OE ablation and OB inactivation lead to prolonged freezing periods in the conditioned fear retrieval paradigm, and the latter treatment also increases baseline freezing.

## Results

### Distinct respiration patterns during fear-induced freezing.
We first examined respiration patterns in freely behaving mice during auditory conditioned fear behaviors. Prior to behavioral experiments, a thermocouple was chronically implanted in the nasal cavity to monitor respiration and bipolar electrodes were implanted in the OB to record LFP activity. The conditioned stimulus was a 10 s pure tone (5 kHz) and the unconditioned stimulus was a 1 s foot shock (0.5 mA) starting at the end of the tone. During conditioning, mice received four tones paired with foot-shocks in a 10 min session. Twenty-four hours later, they were subject to a retrieval test in a novel environment. After a ~3 min free exploring period, the conditioned tone was presented four times within 12 min and mice showed robust freezing

behavior (Fig. 1a). During both non-freezing and freezing periods, thermocouple signals and OB LFPs were highly correlated (Fig. 1b) as the low frequency (2–12 Hz) oscillation of the OB LFP is entrained by nasal airflow[25, 26]. Thus, the breathing rate could be measured directly from thermocouple signals or approximated from OB LFPs. Respiration patterns were markedly different during freezing compared to non-freezing periods as evidenced by power spectrum density estimates from thermocouple and OB LFP signals (Fig. 1b). During pre-tone baseline periods, respiration rates, determined by the peak in the power spectrum of OB LFP or thermocouple recordings, rapidly alternated between sniffing bouts (7–12 Hz) and slow breathing (<4 Hz). In contrast, during freezing periods, respiration rates became very steady with little cycle-to-cycle variability for individual animals (Fig. 1b, c) and ranged from 2 to 6 Hz with an average of ~4 Hz across all animals (Fig. 1d). Consequently, the percent power in the 2–6 Hz band increased significantly during freezing compared to baseline non-freezing periods (Fig. 1e). These results indicate that fear-related behavioral states (e.g., freezing) are characterized by distinct respiratory patterns in mice.

### Correlation between OB and plPFC activity during freezing.
Recent studies on conditioned fear circuitry reveal a predominant 4-Hz oscillation in the plPFC[21, 22], a structure essential for expression of the freezing behavior[27–30]. Since this frequency is similar to the observed breathing rate during freezing, we tested whether these two rhythms are correlated. We repeated the auditory conditioned fear experiments in freely behaving mice with electrodes implanted in both the OB and plPFC. During freezing, oscillations in the simultaneously recorded OB and plPFC LFPs were dominated by a low frequency component at ~4 Hz and strongly coupled (Fig. 2a, b). The phase coherence was greatest in the 2–6 Hz range (peak at ~4 Hz) and the peak coherence increased significantly from non-freezing to freezing periods (Fig. 2c). Similarly, the cross-correlation between the OB and plPFC LFPs increased during freezing (Fig. 2d). Circular distribution analysis of the OB and plPFC phase differences revealed a 13.5° phase difference during freezing (Fig. 2e), suggesting that the plPFC oscillations were not a result of volume conduction from the OB. These findings reveal that the rhythmic activity in the plPFC is coupled to respiration during conditioned fear-induced freezing behavior.

### Contribution of olfactory inputs to plPFC rhythmic activity.
The respiration-locked plPFC rhythm during freezing can potentially arise from several mechanisms, which are not mutually exclusive. It may be intrinsically generated in the PFC[22], influenced by the respiration centers in the brainstem, and/or modulated by respiration-entrained olfactory inputs. Here, we focused on the potential contribution of nasal airflow to the plPFC rhythm during freezing. We used a within-subjects, unilateral naris occlusion approach to block nasal airflow from one side since the ascending olfactory pathway is predominantly ipsilateral. Seven mice were implanted with bilateral electrodes in the OB and plPFC and then underwent permanent unilateral naris occlusion before subsequent fear conditioning. Occlusion blocked airflow, which was evident by decreased respiration-related oscillations in the OB LFPs on the occluded side during the anesthetized state (Fig. 3a, b). Unilateral naris occlusion did not influence freezing behavior presumably due to the intact olfactory inputs from the open side. However, we observed a significant reduction in the coherence and cross-correlation between OB and plPFC activity on the occluded side as compared to the open side (Fig. 3c–e) supporting a functional role of nasal airflow in modulating the plPFC rhythm in the conditioned fear

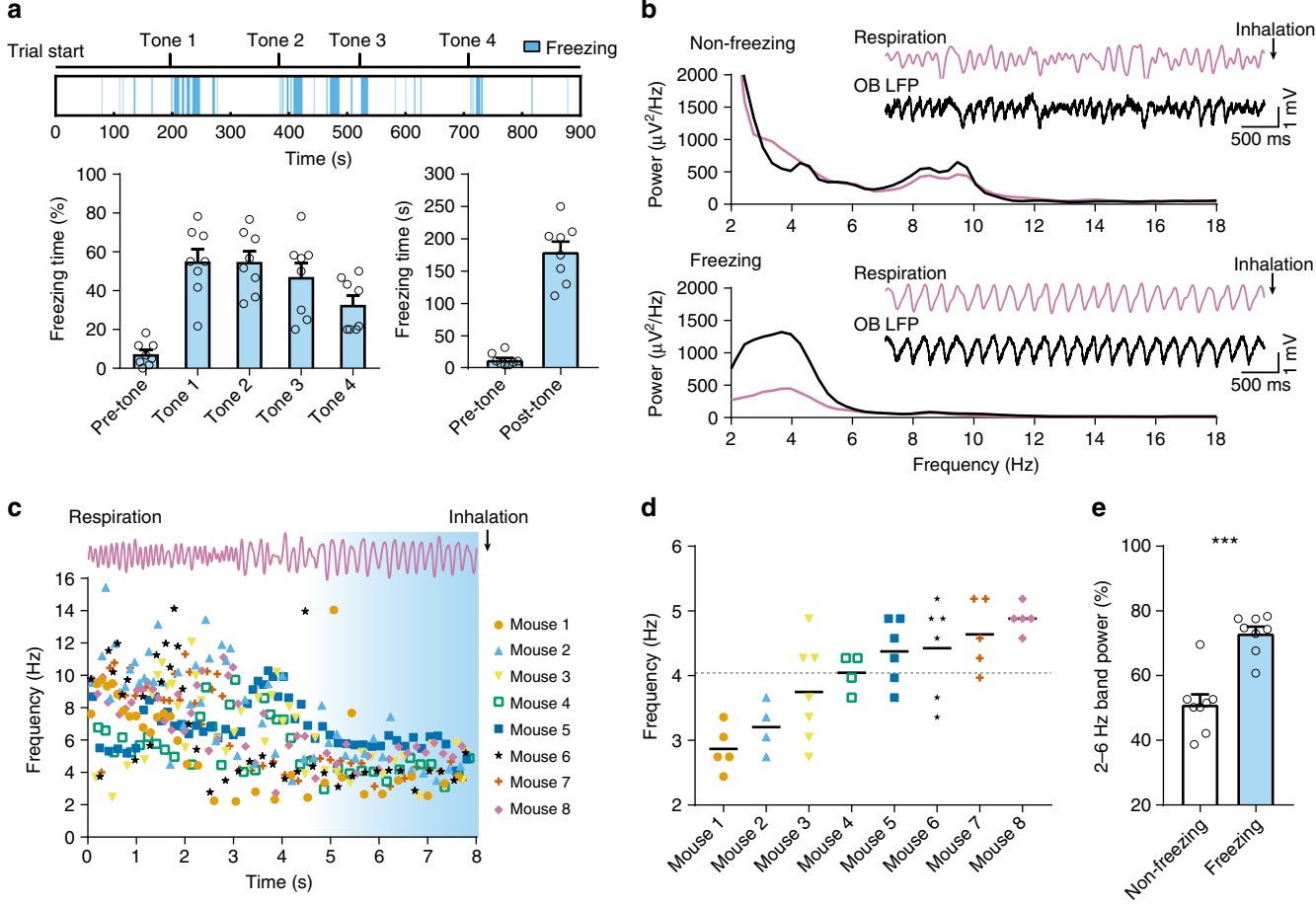

**Fig. 1** During fear retrieval, mice breathe at a characteristic frequency (between 2–6 Hz) while freezing. **a** Top: The full time course of a retrieval session for a single mouse 24 h after fear conditioning to pure tones. Freezing periods are especially prominent after tone presentations, but also happen during inter-stimulus intervals. Bottom left: quantification of freezing behavior for 8 mice expressed as a percentage of time spent freezing in the 60 s period after each tone presentation. The pre-tone freezing is measured in the 60 s period before the first tone. Bottom right: total freezing time in seconds before (pre-tone) and after (post-tone) the first tone onset. **b** Power spectral density of the thermocouple (downward deflections represent inhalation, applying to all respiration traces) and OB LFP recordings from a single mouse during non-freezing (from the pre-tone baseline, top) and freezing (bottom) periods. Insets show simulatneously recorded, unfiltered thermocoulple and OB LFP signals. **c** The instananeous respiratory frequency aligned to a freezing period from each mouse. Top: A raw thermocouple trace from an example mouse (Mouse 8). Bottom: The instantaneous (cycle-to-cycle) frequency of either thermocouple signals or low-pass filtered OB LFPs aligned to the onset of a single freezing period (onset at 5 s). **d** The respiratory frequency of each mouse during continuous freezing episodes longer than 5 s, estimated by the peaks in the power spectrum of either thermocouple signals or low-pass filtered OB LFPs. The overall average for all mice is 4.05 ± 0.13 Hz (indicated by the dashed line). **e** The percent power in the 2–6 Hz band during non-freezing baseline as compared to freezing periods (51.20 ± 3.22 to 73.23 ± 2.13, $n = 8$ paired two-tailed Student's $t$ test $t(7) = 9.67$, $p < 0.001$)

behavior. This approach might underestimate the contribution of the olfactory inputs due to potentially incomplete naris closure, residual respiration-related inputs to the OB in the occluded side via the contralateral AON or other brain regions, and reciprocal inputs between the open and occluded-side plPFC.

**Connectivity between the olfactory pathway and plPFC.** The plPFC integrates information from diverse brain regions involved in a variety of cognitive functions and behaviors[31]. We next investigated how respiration-locked olfactory inputs reach the plPFC using both retrograde and anterograde circuit tracing approaches. The retrograde canine adenovirus type 2 (CAV2)-Cre virus ($13 × 10^{12}$ GC/ml, 0.5 μl) was focally injected into one side of the plPFC in the Rosa26-floxed-tdTomato reporter mice ($n = 3$) and labeling patterns were examined four weeks later. Among the OB and olfactory cortices, which receive direct OB projections, we observed tdTomato$^+$ cell bodies primarily in the ipsilateral AON and TT (Fig. 4a), consistent with previous reports[12, 32, 33]. In a complementary experiment, we injected Cre-

dependent channelrhodopsin-2 (ChR2)-EYFP virus into the AON of Vglut1-Cre mice ($n = 3$) to label glutamatergic pyramidal neurons in the AON. Four weeks later, labeled axonal fibers were found in both the prelimbic (PL) and infralimbic (IL) prefrontal cortex on the ipsilateral side of the injection (Fig. 4b). Notably, the AON/TT is a main target of the OB tufted cells[34], which receive direct synaptic inputs from OSNs in the nose and display a strong respiration-entrained rhythm[11]. To demonstrate that this anatomical pathway is functional, we optogenetically activated OSNs expressing ChR2 under the control of olfactory marker protein (OMP) gene at 13 Hz, a frequency distinct from respiration. As expected, activation of OSNs entrained LFP responses in both the OB and plPFC (Fig. 4c, d). These data provide support for a previously underappreciated neural connection that links the olfactory system with the plPFC.

**Loss of olfactory inputs leads to prolonged freezing.** To investigate the consequences of disrupting olfactory inputs on the fear circuit and behavior, we first used methimazole to lesion the

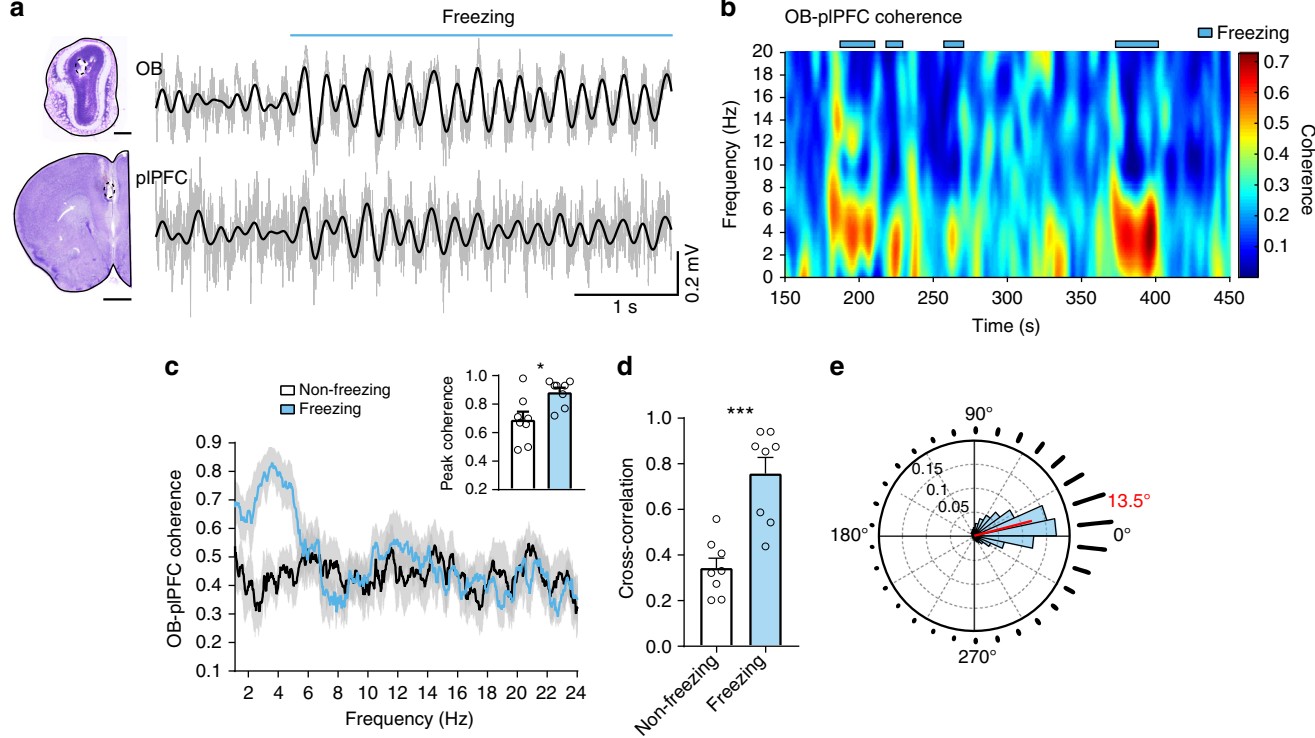

**Fig. 2** During freezing, the neural activity in the OB and plPFC is dominated by highly correlated ~4-Hz oscillations. **a** Example OB and plPFC LFPs during a freezing epoch. Filtered signals (2–6 Hz) are overlaid on raw traces (gray). Nissl-stained coronal brain sections show electrode sites. Scale bars, 0.5 mm top and 1.0 mm bottom. **b** Time-frequency coherogram of OB and plPFC LFPs during a portion of a retrieval session in which freezing was observed (marked in blue). High phase coherence emerges during periods of freezing. **c** Spectral coherence between simultaneously recorded OB and plPFC LFPs during non-freezing (from pre-tone baseline, black) and freezing (blue) periods (mean ± SEM, $n = 8$ mice). Inset, averaged peak coherence for non-freezing and freezing periods ($0.69 \pm 0.06$ to $0.88 \pm 0.03$, $n = 8$, Wilcoxon matched-pairs signed rank test, $p = 0.016$). **d** All animals tested showed an increase in the maximum OB-plPFC cross-correlation value from non-freezing to freezing periods ($0.34 \pm 0.04$ to $0.75 \pm 0.07$, $n = 8$, paired two-tailed Student's $t$ test $t(7) = 5.75$, $p < 0.001$). The LFPs were filtered at 2–6 Hz. **e** Circular distribution of phase differences between OB and plPFC 2–6 Hz signals during freezing. The mean direction of the distribution is 13.5° (red line, with lower 95% confidence limit = 13.0 and upper 95% confidence limit = 14.0, $p < 0.001$ in one-sample test for mean angle equal to 0°). The phase difference distribution is also visualized as a radial histogram (30 bins) surrounding the polar plot

nasal epithelium for reasons detailed in the Discussion. Four days following intraperitoneal injection of 75 mg/kg methimazole, OSNs were essentially absent from the OE (Fig. 5a). Tissue sections were double stained by antibodies against OMP (labeling mature olfactory sensory neurons, red) and SUS-4 (labeling supporting cells and Bowman's gland and duct cells, green). Elimination of OSNs significantly reduced the coupling between the OB LFPs and respiration in both anesthetized and awake mice, but did not completely remove it (Fig. 5b, c). Note that the peak coherence between respiratory and OB signals always occurred at the dominant breathing frequency, but for shuffled data the peak could occur randomly in the full frequency range (between 1–12 Hz). The data were collected from a total of 17 animals and each condition was tested in a subset of animals (control awake $n = 7$, methimazole awake $n = 5$, control anesthetized $n = 5$ and methimazole anesthetized $n = 5$). Residual coupling, more evident in the awake state, may be due to respiration-related inputs from other brain regions to the OB or potential regrowth of sensory inputs, which would be minimal at this time point[35]. Note that methimazole treatment did not change the breathing rates during normal behaviors (Fig. 5d; $F(1,3) = 0.6$, $p = 0.495$ for pre vs post condition in two-way ANOVA with repeated measures).

A separate group of methimazole treated mice ($n = 9$) underwent the same auditory fear conditioning and retrieval paradigm four days post injection. We found that removing peripheral olfactory inputs reduced the 4-Hz oscillation in the plPFC, but did

not completely eliminate it. The coherence between OB and plPFC LFPs was generally low and did not significantly increase during freezing (Fig. 5e, f), but a peak at the respiration rate (~4 Hz) was still evident (Fig. 5f). However, the peak coherence during freezing was significantly reduced compared to control animals (methimazole treated $0.59 \pm 0.04$, $n = 9$ vs control $0.88 \pm 0.03$, $n = 8$; U = 1.5, $p = 0.0002$ in two-tailed Mann–Whitney test; c.f Fig. 5f to Fig. 2c). Consistently, the cross-correlation between OB and plPFC activity in methimazole treated mice did not increase during freezing (c.f. Fig. 5g to Fig. 2d) and circular distribution analysis failed to reveal a dominant phase difference as observed in control animals (c.f. Fig. 5h to Fig. 2e).

Behaviorally, methimazole treated mice still showed freezing behavior during the retrieval trials (Fig. 6a, b), indicating that olfactory inputs are not required for acquisition or expression of conditioned fear behavior. Remarkably, these animals had significantly longer freezing periods than control as well as unilaterally naris-occluded animals over the course of the retrieval session (Fig. 6b, c). This was not due to non-specific decreases in locomotion after methimazole treatment since the pre-tone baseline freezing time was not significantly different among the three groups (Fig. 6a). Thus, this result suggests that lacking olfactory inputs increases freezing time during retrieval.

Since methimazole treatment may have other non-olfactory off-target effects, we used another method to further confirm that disruption of olfactory signals could prolong freezing periods. We bilaterally inactivated the OBs of cannulated mice with

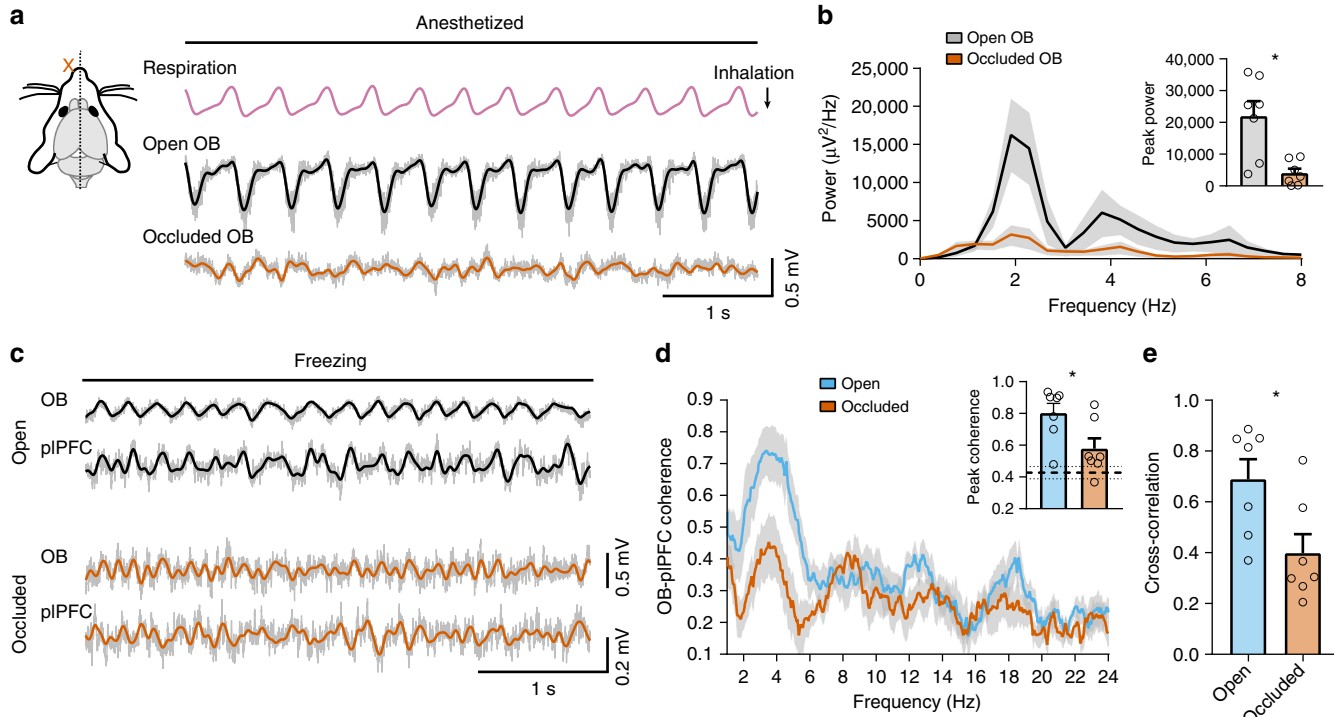

**Fig. 3** Respiration-entrained olfactory signals contribute to the 4-Hz rhythm in the plPFC. **a** Unilateral naris occlusion impairs ascending signals from the olfactory system. Occlusion could be confirmed by the reduction of respiration-entrained oscillations in the OB, especially in the anesthetized state. Filtered signals (2–6 Hz) are overlaid on raw traces (gray), which also applies to (**c**). **b** Power spectral density of OB LFPs from open and occluded sides under anesthesia. Inset, averaged peak power of the OBs from open vs occluded sides (21987 ± 4716 vs 4993 ± 1452, $n = 7$, paired two-tailed Student's $t$ test $t(6)$ = 3.662, $p = 0.011$). Note that the breathing rate was at ~2 Hz under anesthesia. **c** Bilateral OB and plPFC LFP recordings from a naris-occluded mouse during freezing in the fear retrieval session. **d** Spectral coherence between simultaneously recorded OB and plPFC LFPs during freezing periods on the open vs occluded sides (mean ± SEM, $n = 7$ mice). Inset, averaged peak coherence between the OB and plPFC LFPs for open and closed sides (0.80 ± 0. 06 to 0.58 ± 0.07, $n = 7$, paired two-tailed Student's $t$ test t(6) = 3.211, $p = 0.018$). The average peak coherence between plPFC and shuffled OB signals from the occluded side is 0.42, indicated by the thick dashed line (the two thin dashed lines indicate the standard errors). **e** The occluded side shows a decrease in the OB-plPFC cross-correlation compared to the open side during freezing (0.69 ± 0.08 open vs 0.40 ± 0.08 occluded, $n = 7$, paired two-tailed Student's $t$ test $t(6) = 3.353$, $p = 0.015$)

tetrodotoxin (TTX, 0.5 μl, 60 μM) immediately before the fear retrieval test. Infusion of TTX caused a dramatic loss of neural activity in the OBs (Fig. 6d) and significantly longer freezing periods as compared with saline infused controls (Fig. 6f), consistent with methimazole treatment. Interestingly, acute inactivation of the OB activity also increased the baseline freezing time (Fig. 6e), suggesting that disruption of the OB may have more severe effects than disruption of the OE. The latter treatment spares intrinsic oscillations and centrifugal inputs and leaves residual respiration-entrained activity in the OB (Fig. 5c). Curiously, TTX infused mice did not change the total distance traveled during the pre-tone period (Fig. 6h), albeit with more freezing time (Fig. 6e), suggesting that they moved at a faster speed. This is consistent with previous reports that bulbectomized rodents show altered exploratory behavior with hyperactivity[36, 37].

## Discussion

It has been reported that during conditioned fear-induced freezing behavior, the plPFC exhibits a strong ~4-Hz oscillation, which coincides with freezing epochs[21, 22]. Here we reveal that this oscillation is correlated with the respiration rhythm and OB activity (Figs. 1, 2). Furthermore, unilateral naris closure or pharmacological ablation of the OE significantly reduces this oscillation, suggesting that olfactory inputs contribute to the plPFC activity (Figs. 3 and 5). The mechanosensitive OSNs in the

nose[7], presumably activated by rhythmic nasal airflow[8], entrain high-amplitude OB oscillations with respiration (Fig. 1). Notably, there are two distinct efferent pathways arising from the OB mitral and tufted cells, respectively[34]. The AON/TT is a major target of tufted cells, which may be primed to carry airflow information, since they respond early in the inhalation phase and have a lower threshold for spike response to olfactory nerve input compared to mitral cells[11, 38, 39]. Our retrograde and anterograde tracing experiments confirm the OB→AON/TT→plPFC pathway. Consistent with this anatomical connection, optogenetic activation of OSNs entrains the plPFC activity (Fig. 4). It remains possible that alternative pathways carrying olfactory information[40] also contribute to the OSN-related plPFC activity.

The strong correlation between respiration and plPFC rhythmic activity during freezing behavior has important implications on the significance of this 4-Hz oscillation. One possibility is that respiration-related OB inputs modulate rhythmic activity of multiple sites including plPFC and amygdala in the fear circuit (Fig. 6i) and may facilitate or synchronize network oscillations for behavioral output[18], as previously proposed by others[22]. It is worth noting that the 4-Hz oscillation in the plPFC may not solely rely on olfactory inputs as unilateral naris closure or ablation of the OE reduces the power at this frequency, but does not completely eliminate it (Figs. 3, 5). The remaining 4-Hz oscillation in the plPFC could be attributed to the residual respiration rhythm in the OB, intrinsically generated, and/or originated from other brain regions. Our study also raises another

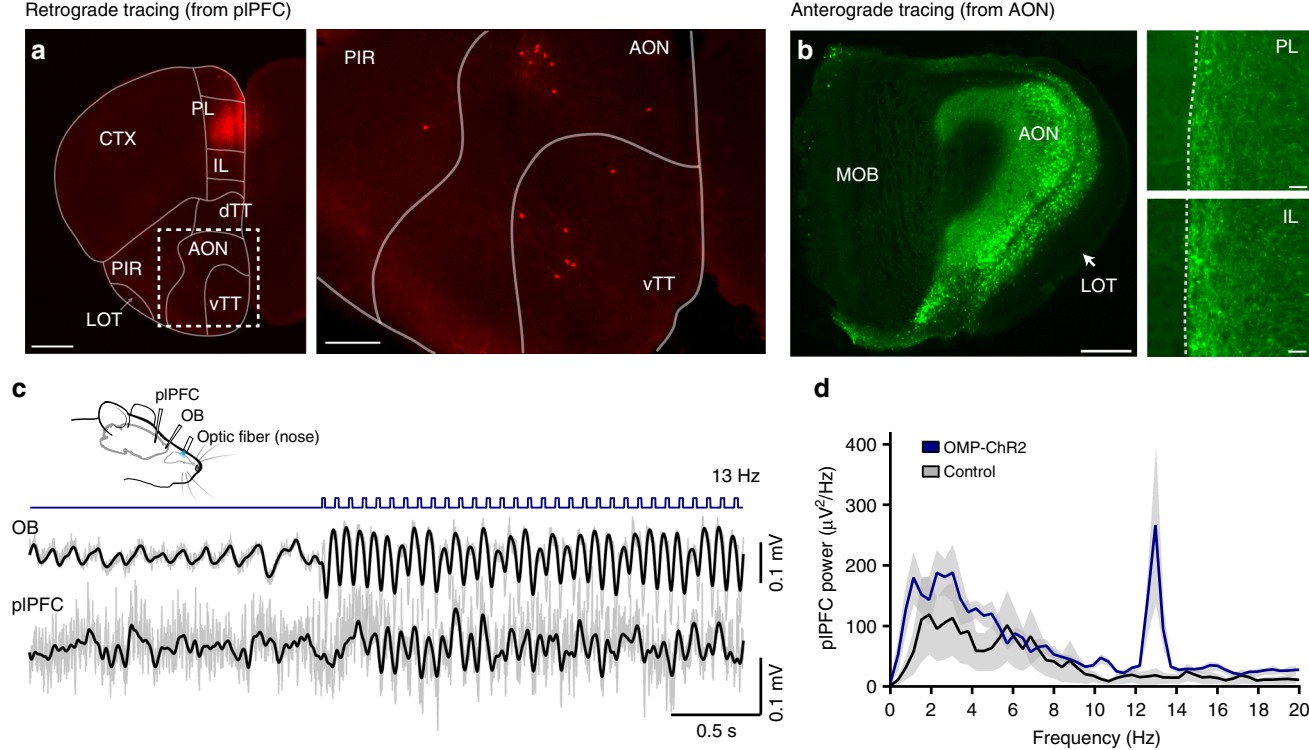

Retrograde tracing (from plPFC)

Anterograde tracing (from AON)

**Fig. 4** Optogenetic stimulation of olfactory sensory neurons in the nose entrains rhythmic activity in the OB and plPFC. **a** The prelimbic PFC (PL) receives direct inputs from the ipsilateral anterior olfactory nucleus (AON) and ventral taenia tecta (vTT). Focal injection of 0.5 μl CAV2-cre (13 × 10^12 GC/ml) into the PL in *Rosa^tdTomatof/f* reporter mice leads to labeled cell bodies in the AON and vTT, two regions receiving direct inputs from the OB. CTX cortex, PIR piriform cortex, LOT lateral olfactory tract, dTT dorsal taenia tecta. Scale bar: 750 μm, left panel and 250 μm, right panel. **b** Focal injection of Cre-dependent ChR2-EYFP virus directed to the AON in Vglut1-Cre mice results in labeled fibers in both the prelimbic (PL) and infralimbic (IL) PFC. Note that there is essentially no viral infection in the main olfactory bulb (MOB, left panel) and the projection from AON to the prefrontal cortex is only on the ipsilateral side (right panel). Scale bar: 200 μm, left panel and 30 μm, right panel. The dotted lines mark the midline separating the two hemispheres. **c** An optical fiber was implanted in the nose of OMP-ChR2 mice to stimulate OSNs. The example shows stimulation at 13 Hz (15 mW, 20 ms), which is distinct from endogenous respiration rhythms. Simultaneous LFP recordings from the OB and plPFC show optical stimulation-entrained neural activity. Filtered signals (2–20 Hz) are overlaid on raw traces (gray). Blue bars mark the laser (473 nm) pulses. **d** The power spectral density (mean ± SEM, $n = 3$ mice) of plPFC LFPs shows increased power at 13 Hz, the optical stimulation frequency (blue). The black line shows data from control mice ($n = 3$, $OMP^{Cre/WT}$ without ChR2) that underwent the same procedure as OMP-ChR2 mice

possibility that the 4-Hz oscillation may be an epiphenomenon of the respiratory pattern during the freezing behavior. In fact, the power of the 4-Hz oscillation in the plPFC is not positively correlated with freezing duration: disruption of peripheral olfactory inputs reduces the power of the 4-Hz oscillations in the plPFC (Fig. 5), but increases freezing time (Fig. 6). Further studies that manipulate specific pathways in different behavioral contexts are required to distinguish these possibilities and establish the functional significance of respiration-related neural rhythms[41].

In this study, we acutely removed bilateral olfactory signals by ablating the nasal epithelium or reversibly inactivating the OB instead of permanent or chronic disruption for several reasons. Primarily, permanent or chronic removal of olfactory inputs has profound effects on emotion and behavior. In rodents, surgical removal of the OBs leads to anxiety and depression-like behaviors that can be reversed by antidepressant treatment[36, 37]. Likewise, genetically modified mouse lines with functional ablation of the main olfactory system display increased anxiety-like behaviors[42, 43]. In addition, inducible knockout of key olfactory signaling molecules in OSNs is likely compromised by the ongoing neurogenesis in the OE. Ablation of the OE via methimazole treatment is complete and does not change breathing rates or the baseline freezing time, but it increases the total freezing time during conditioned fear retrieval trials (Figs. 5, 6).

Our study does not completely rule out the potential contribution of off-target effects of this treatment to the behavioral change. Nevertheless, methimazole treatment seems to have less severe effects than disruption of the OB activity via TTX infusion. The latter treatment not only prolongs retrieval freezing time, but also increases the baseline freezing time (Fig. 6). Although both manipulations disrupt olfactory inputs, they may act on partially independent mechanisms. Unlike OB TTX infusion, methimazole treatment spares other centrifugal and non-olfactory inputs and intrinsic oscillations in the OB. It remains to be determined whether and how the olfactory-plPFC pathway contributes to olfactory loss-induced, prolonged freezing periods, given that rodents heavily rely on olfactory cues in normal behaviors.

Our study demonstrates that mice show a distinct breathing pattern during conditioned fear-induced freezing behavior (Fig. 1). This finding is consistent with the tight emotion-respiration link observed across species including rodents and humans[44]. In addition to autonomic and emotional regulation of breathing rhythms, humans can change respiration patterns in a voluntary and conscious manner as practiced in yoga and meditation[45–47]. Either involuntary or voluntary changes in nasal breathing can potentially modulate brain activity through the neural circuits between olfactory and emotional centers including the plPFC studied here (Fig. 6i). Indeed, nasal (but not oral) breathing is effective in entraining the neural activity in the limbic

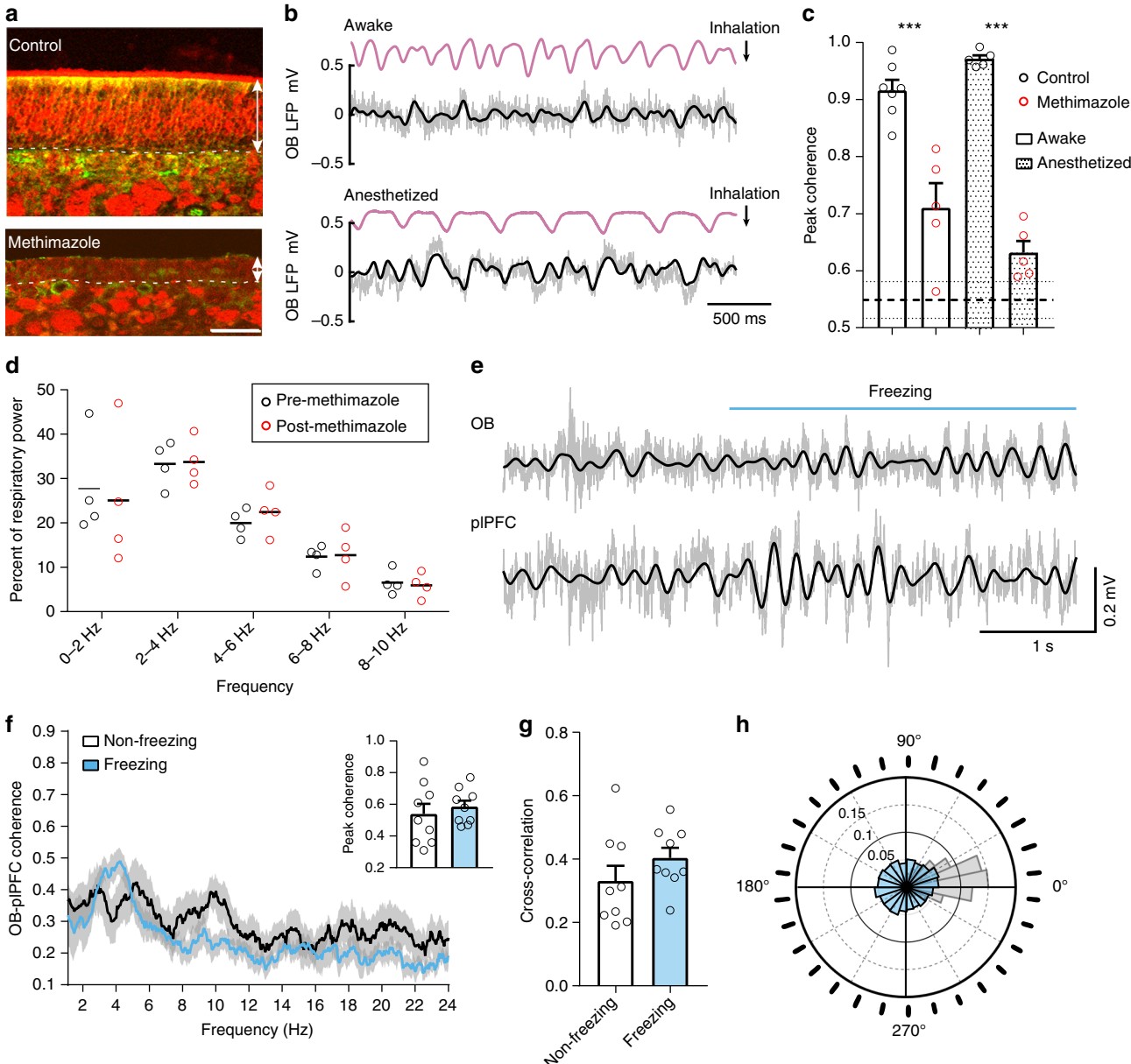

**Fig. 5** Disruption of olfactory inputs decouples OB LFPs from respiration during freezing. **a** Sections of olfactory epithelia from six-week-old mice. Arrow lines denote the thickness of the olfactory epithelium. Dashed lines mark the basement membrane that separates the olfactory epithelium from the underneath lamina propria. Asterisks mark OMP[+] axonal bundles. Scale bar: 50 μm. **b** Thermocouple and OB LFP recordings from a single mouse following methimazole injection. The OB oscillations no longer faithfully follow respiration, in contrast to control condition (c.f. Fig. 1b top). **c** Average peak coherence between respiratory signals and OB LFPs (one-way ANOVA $F(3,17) = 35.1$, Tukey post hoc, awake control vs methimazole, $p < 0.001$; anesthetized control vs methimazole, $p < 0.001$). The average peak coherence between respiration and shuffled OB signals is 0.55, indicated by the thick dashed line (the two thin dashed lines indicate the standard errors). **d** The percent of power in 5 frequency bands of respiration recorded in awake, behaving mice ($n = 4$) is unchanged following methimazole treatment. Respiratory signals (2 hr) were recorded pre and post methimazole injection in the home cage and analyzed in 10 s epochs. **e** Example OB and plPFC LFPs during a freezing epoch in a methimazole treated mouse. Filtered signals (2–6 Hz) are overlaid on raw traces (gray). **f** Spectral coherence between simultaneously recorded OB and plPFC LFPs during non-freezing baseline (black) and freezing (blue) periods (mean ± SEM, $n = 9$ mice). Inset, averaged peak coherence for non-freezing and freezing periods (0.54 ± 0.06 to 0.58 ± 0.04, $n = 9$, paired two-tailed Student's $t$ test $t(8) = 0.552$, $p = 0.596$). **g** Maximum OB-plPFC cross-correlation values during non-freezing and freezing periods (0.33 ± 0. 05 to 0.40 ± 0.03, $n = 9$, paired two-tailed Student's $t$ test $t(8) = 1.078$, $p = 0.313$). **h** Circular distribution of phase differences between OB and plPFC 2–6 Hz signals during freezing (Rayleigh test for circular non uniformity $z = 2.08$, $p = 0.125$). Data from methimazole treated animals are overlaid on the distribution from control animals (in gray as in Fig. 2e)

system of human brain[48], even though humans breathe at a much slower rate (0.1–0.3 Hz) than rodents (2–12 Hz). It would be interesting to further dissect out the neural mechanisms underlying the mental and behavioral benefits of self-regulated respiration practices.

## Methods

**Animals.** Wild-type C57BL6/J mice were purchased from the Jackson Laboratory. *OMP-ChR2* (*OMP^{Cre/WT}-Rosa^{ChR2f/f}*) mice were obtained by crossing *OMP-Cre* mice (the coding region and part of the 3′ UTR of OMP was replaced by that of Cre; JAX Stock No: 006668)[49] and *Rosa26-CAG-LSL-ChR2(H134R)-EYFP-WPRE* mice (JAX Stock No: 024109 or Ai32 line)[50]. *Rosa^{tdTomatof/f}* mice (*Rosa26-CAG-*

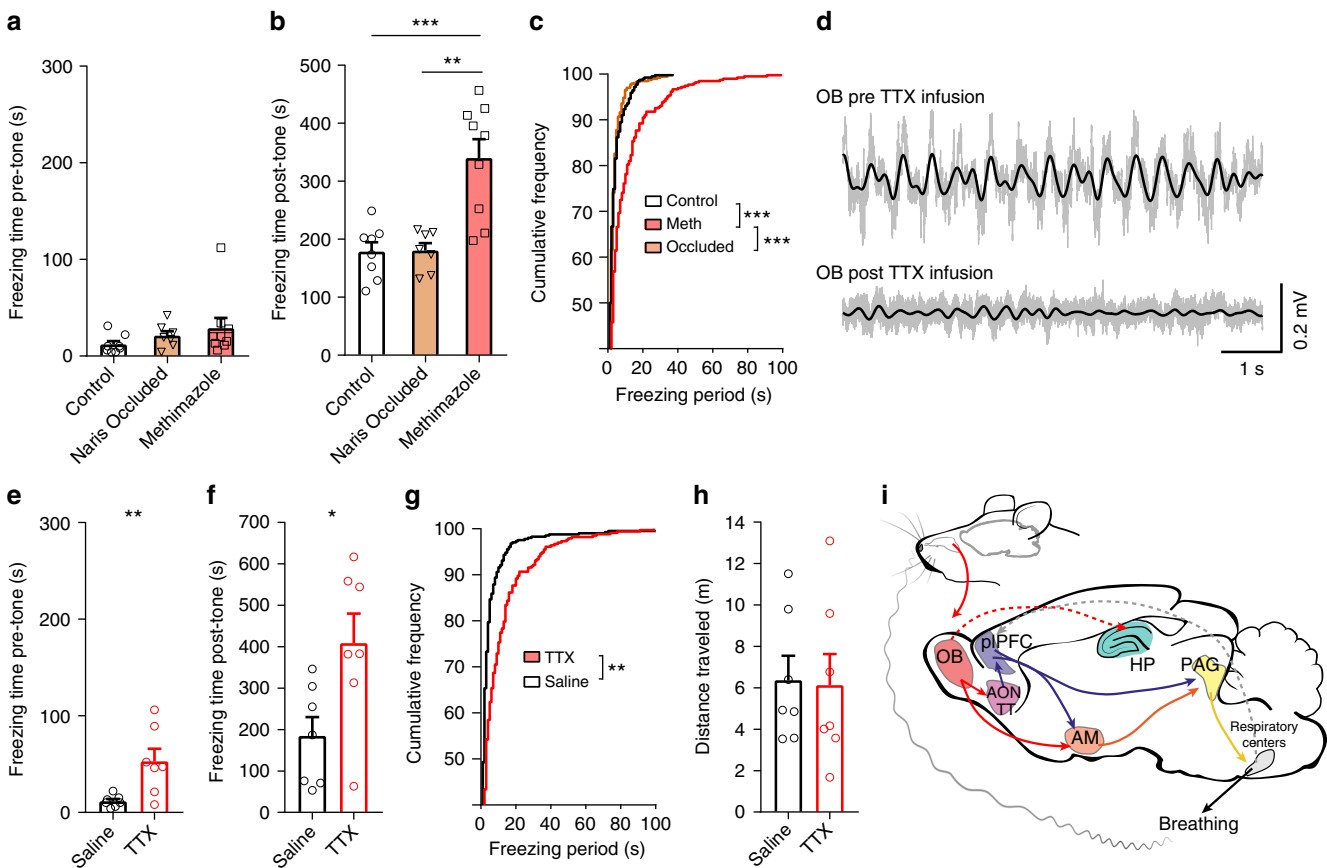

**Fig. 6** Disruption of olfactory inputs prolongs freezing periods. **a** Baseline (pre-tone) freezing time is not significantly different among control ($n = 8$), unilateral naris occlusion ($n = 7$) and methimazole treated ($n = 9$) groups (one-way ANOVA $F_{(2,21)} = 1.281$, $p = 0.299$). **b** Freezing time during the retrieval trial is significantly longer in methimaole treated mice compared to control and naris-occluded animals (one-way ANOVA $F_{(2,21)} = 15.61$, Tukey post hoc, control vs methimazole $p < 0.001$, naris occluded vs methimazole $p < 0.001$). **c** Cumulative frequency distributions show methimazole treated animals have an increased frequency of longer freezing periods compared to control or naris-occluded mice ($p < 0.001$, two-sample Kolmogorov–Smirnov test). **d** Example traces of OB LFPs, 2–6 Hz filtered traces are overlaid on raw traces (gray) before (pre TTX) and 30 min after intrabulbar injection of TTX (post TTX). **e** Baseline (pre-tone) freezing time is significantly longer in TTX infused mice ($53.71 \pm 13.13$ s, $n = 7$) compared to saline infused mice ($n = 7$) ($12.43 \pm 2.27$ s, $n = 7$; unpaired two-tailed Student's $t$ test $t(12) = 3.099$, $p = 0.009$). **f** Freezing time is significantly longer in TTX infused mice ($410.60 \pm 71.39$ s) compared to saline infused mice ($186.60 \pm 45.59$ s; unpaired two-tailed Student's $t$ test $t(12) = 2.645$, $p = 0.021$). **g** Cumulative frequency distributions show TTX infused animals have an increased frequency of longer freezing periods compared to saline infused mice ($p = 0.01$ in two-sample Kolmogorov–Smirnov test). **h** Total distance traveled prior to the tone onset is not different in saline ($6.40 \pm 1.18$ m) and TTX infused mice ($6.16 \pm 1.51$ m; unpaired two-tailed Student's $t$ test $t(12) = 0.127$, $p = 0.901$). **i** Entrained olfactory inputs modulate the rhythmic activity of the conditioned fear circuit. Solid and dotted lines denote direct and indirect connections, respectively. Note that many connections are reciprocal, which are not shown for simplicity. AM amygdala, HP hippocampus, PAG periaqueductal gray

*LSL-tdTomato-WPRE* or Ai9 line, JAX Stock No: 007905[50] were used as the reporter line in the retrograde tracing experiments, and *Vglut1-Cre* mice (*Vglut1-IRES2-Cre-D*; JAX Stock No: 023527)[51] were used in the anterograde tracing experiments. All mice were kept on a 12 h light/12 h dark cycle, provided with food and water ad libitum, and housed individually after surgery. All behavioral and recording procedures were performed during the light phase of the cycle. Both male and female mice (2–6 months old, unless otherwise stated) were used and the data were combined since no sex differences were evident. Mice were randomly allocated into control or experimental groups. All procedures were approved by the University of Pennsylvania Institutional Care and Use Committee.

**Surgical implantation**. Mice were exposed to isoflurane at 3% (vol/vol) for anesthetic induction. They were then secured in a stereotaxic system (Model 940, David Kopf Instruments) and isoflurane levels were reduced to 1.5% for the remainder of the surgery. Body temperature was maintained at 37 °C with a temperature control system (TC-1000, CWE). Areas for electrode implantation were measured from bregma using a stereotaxic arm and standard coordinates[52]. Craniotomies were made over the OB (+4.3 mm AP, +1.0 mm ML, 1.5 mm DV) and plPFC (+1.8–2.0 mm AP, +0.3 mm ML, 1.4 mm DV). To monitor respiratory behavior, some animals also had a hole drilled at the anterior nasal bone for thermistor implantation[53] (36 ga, PFA insulated, Omega Engineering). Thermocouples were implanted in the nasal cavity contralateral to electrode implants. For

optogenetic experiments a fiber optic cannula (400 μm core, 1 mm fiber length, Thor Labs) was implanted in this hole in the nasal cavity. Electrodes consisted of pairs of PFA-coated tungsten wires (50 μm bare diameter, AM-systems) separated by 0.5–1.0 mm in the dorsal-ventral plane. Electrode impedance was measured using an IMP-2AMC (BAK Electronics) and ranged from 50–180 KΩ. Wires were attached to an 18 pin interface board (EIB). Electrodes were referenced to a low impedance stainless-steel wire implanted in the cerebellum and a stainless-steel ground electrode was connected to a cerebellar skull screw. Implants (electrode and thermocouples) were secured using Kwik-Sil adhesive (World Precision Instruments, WPI) and dental cement. In a subset of animals electrodes were bilaterally implanted and one naris was closed by touching a small vessel cauterizer (Fine Science Tools) to the external naris for ~5 s. The occluded side (left vs right) was counterbalanced across animals. Naris occlusion was confirmed by visual examination and by loss of nasal airflow entrained OB LFPs especially under anesthesia (Fig. 3a). After surgery mice were allowed to recover for 7 days. Another subset of mice received a single intraperitoneal injection of methimazole (Sigma, 75 mg/kg in sterile saline) four days prior to fear conditioning and/or recording.

For pharmacology experiments, wild-type mice had 26 gauge guide cannulae (WPI) implanted bilaterally in their OBs. After a one week recovery these mice were fear conditioned and tested the following day with infusion. TTX (60 μM, Abcam, $n = 7$) or sterile saline ($n = 7$) were injected via a 5 μl Hamilton syringe at a rate of 0.1 μl per min using an automated micro-pump. Following injection of 0.5 μl TTX or 0.5 μl saline per bulb, mice were left in their home cage for 40 min before

being transferred to a novel environment for the fear retrieval test (see below). Additional mice (not included in the behavioral group) were implanted with bipolar electrodes in the OB that extended 1 mm past the end of the guide cannula to acquire electrophysiological recordings before and after TTX infusion (Fig. 6d).

**Neurotracing**. For retrograde tracing experiments $Rosa^{tdTomatof/f}$ mice were injected with *CAV2-Cre* virus at a concentration of $13 \times 10^{12}$ GC/ml[54, 55]. Unilateral viral injections were made with a 33 gauge needle (Hamilton) and an Ultra Micro Pump (WPI) mounted to a stereotaxic arm. The needle was lowered to the plPFC (+1.8–2.0 mm AP,+0.3 mm ML, 1.4 mm ventral to the cortical surface). For anterograde tracing experiments *Vglut1-Cre* mice were unilaterally injected with 0.5–0.8 µl Cre-dependent ChR2-EYFP virus (*AAV1.EF1a.DIO-.hChR2(H134R)-eYFP.WPRE.hGH* from Penn Vector Core). The microinjection needle was angled to reach the AON by passing through the OB and reaching the coordinates +3.0 mm AP, +0.9 mm ML, 3.5 mm DV. Virus was infused at a rate of 50 nl/min. Following infusion, the needle was allowed to settle for 15 min to allow viral diffusion before needle retraction. Four weeks following viral injection, the mice were transcardially perfused with PBS and 4% paraformaldehyde (PFA) and brains were dissected. After 4 h fixation at 4 °C, 75 µm coronal vibratome sections were cut, mounted, and visualized under a fluorescent microscope (Leica DM5000 B). Outlines of brain regions (Fig. 4a) were determined using matching coronal sections from the Allen Mouse Brain Reference Atlas[56].

**Optogenetic stimulation**. OMP-ChR2 mice with optic fibers implanted in their nasal cavities were stimulated in their home cage using a 473 nm laser (SLOC Lasers, BL473T8-150FC) coupled to an articulated rotary joint patch cable (Thor Labs). Laser light power was adjusted to achieve an intensity of ~15 mW at the fiber tip (continuous output) and the laser was controlled via delivery of 0–5 V TTL pulses generated by an arbitrary waveform generator (Agilent, 33201 A).

**Fear conditioning**. Fear conditioning was performed in a Med Associates modular test chamber by pairing a 5 kHz pure tone (10 s) with an aversive stimulus (1 s foot shock, 0.5 mA, 4 CS$^+$–US pairings; inter-trial intervals, 120–180 s with the onset of the US coinciding with the offset of the CS$^+$). After 24 h, the conditioned mice were submitted to a 15 min testing session (fear retrieval) in a novel chamber (7½" x 11 ½" x 5"). The mice were connected to the recording apparatus and left in their home cage for 1 h prior to being moved to the novel context and the initiation of a testing session. Freezing behavior was scored manually by visual inspection of recorded videos and animals were considered to be freezing if no movement was detected for 1 s by an experimenter blind to condition. In cases where drugs or saline were used prior to fear retrieval, distance traveled in the first two minutes of the retrieval test (prior to the first tone onset) was determined using behavioral tracking software (ANY-maze behavioral tracing software).

**Data acquisition**. Electrophysiological recordings were made by connecting EIB interface boards to an Intan RHD2000 amplifier board. LFP signals from each electrode were amplified, filtered between 0.1 Hz to 9 kHz, digitized at 25 kHz, and stored on a PC for offline analysis. At the conclusion of the experiment, animals were deeply anesthetized with ketamine/xylazine (200/20 mg/kg body weight). In some cases, recordings were taken from anesthetized animals prior to electrolytic lesioning (~1 mA for 20 s) of recording sites. Under anesthesia, direct measurements of chest expansion using a piezoelectric belt (Kent Scientific) wrapped around the mouse's body were used to measure respiratory rate. Mice were transcardially perfused with PBS and 4% paraformaldehyde and brains were dissected. After overnight fixation, 75 µm coronal vibratome sections were cut, mounted, and stained with cresyl violet to confirm recording sites (Fig. 2a).

**Data analysis**. Acquired data were processed offline using custom Matlab scripts and the multitaper spectral methods included in the Chronux toolbox[57]. Differential LFPs were calculated as the difference between pairs of neighboring electrodes in the same brain area to achieve spatially local measurements of electrical activities and minimize far-field contamination[58]. Analyses were performed during long, continuous freezing periods (8 control mice, 116 s, 9 methimazole treated mice, 180 s, 7 naris-occluded animals, 107 s) and during equivalent length of non-freezing epochs taken before the first tone presentation. Respiratory frequency during long freezing episodes was estimated as the peak frequency of the power spectrum using Welch's method (the Matlab function *pwelch*, 2 s Hamming windows, 0.5 s overlap). Coherence spectra were estimated using the multitaper method implemented in Chronux using bandwidths of 0.5 Hz. Coherence spectra were estimated between thermocouple signals and shuffled OB LFPs or between plPFC and shuffled OB LFPs by rearranging 250 ms bins of data to generate randomized OB signals. Based on these computed coherence spectra we used the peak coherence values in the 1–12 Hz range covering the breathing frequencies under all conditions. To compute phase differences and cross-correlations between LFP signals, raw LFPs were first bandpass filtered using zero-phase distortion digital filters (the Matlab function *filtfilt*). Instantaneous phase of the signals at each time point was estimated using the Hilbert transform. The resulting phase difference distributions were analyzed using the circular statistics toolbox in Matlab. Cross-correlations between pairs of simultaneously recorded neural activities were calculated using the *xcorr* function

with 'coeff' option. Statistical analysis was performed in GraphPad Prizm 7 (San Diego, CA). Parametric tests were used when the data were normally distributed (Shapiro–Wilk normality tests) with equal variance between groups. If data were not normally distributed (e.g., Fig. 2c inset) the non-parametric Wilcoxon matched-pairs signed rank test or Mann–Whitney U test was used. Data from completed experiments were all included. The experimental sample sizes were adequately powered (at 0.9) to observe the effects at the significance level of 0.05 following the 'Guidelines for the Care and Use of Mammals in Neuroscience and Behavioral Research' (https://www.ncbi.nlm.nih.gov/books/NBK43321/).

**Immunohistochemistry**. Mice were deeply anesthetized by intraperitoneal injection of ketamine/xylazine (200 mg/20 mg/kg body weight) before decapitation. The heads were fixed in 4% paraformaldehyde (Sigma) overnight at 4 °C. The tissues were then decalcified in 0.5 M EDTA (pH 8.0, ethylenediaminetetraacetic acid) for four days and infiltrated in a series of sucrose solutions before being embedded in OCT. The frozen tissues were cut into 20 µm coronal sections on a cryostat. After antigen retrieval in a 95 °C water bath for 10 min, the tissue sections were blocked for 60 min in 0.3% Triton X-100 in PBS with 3% bovine serum albumin, and then incubated at 4 °C with the primary antibodies in the same solution overnight. The primary antibodies included chicken anti-OMP[59] (1:500, a generous gift from Dr. Qizhi Gong, University of California Davis) and mouse monoclonal SUS-4[60] (1:100, a generous gift from Dr. James Schwob, Tufts University). Immunofluorescence was achieved by reaction with appropriate secondary antibodies (from Molecular Probes, Invitrogen) goat anti-chicken-568 (A11041) and donkey anti-mouse-488 (A21202) at 1:200 for one hour. Tissues were washed in 0.3% Triton X-100 in PBS and mounted in Vectashield (Vector Laboratories). Fluorescent images were taken under a SP5/Leica confocal microscope with LAS AF Lite software.

**Data availability**. All relevant data are available from the authors.

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

## Acknowledgements

We thank Shaohua Zhao, Wenbin Yin, William Olson, and Kim Kridsada for their technical assistance in the immunostaining and neurotracing experiments and Ming Lu for his advice on statistical analysis. This work was supported by grants from the National Institute of Deafness and Other Communication Disorders, National Institute of Health (R01DC006213), and the National Science Foundation (1515930).

## Author contributions

A.H.M. and M.M. conceived the project, A.H.M., W.L., and M.M. designed the experiments and wrote the paper, A.H.M., M.S., and J.P.B. conducted the experiments and analyzed the data, and L.S.Z. provided the *CAV2-Cre* virus.

## Additional information

**Competing interests:** The authors declare no competing interests.

