## [Peer Review File · Nature Communications]

PEER REVIEW FILE

Reviewers' comments:

Reviewer #1 (Remarks to the Author):

This study addresses respiratory coupling between the OB and prelimbic PFC. The topic is important and timely, given the recent small proliferation of studies in mice and humans showing that there is respiratory coupling between the olfactory system and hippocampus and other cortical areas. The data suggest that respiratory activity modulates freezing behavior. Since recent studies have suggested that 4 Hz stimulation of the pPFC is sufficient to induce freezing, the current report is addressing where that 4 Hz input might come from. They show that mice breathe at 4 Hz when they freeze and that this 4 Hz signal is present in both the OB and pPFC with apparently zero phase. The authors do a nice job of linking this rhythmic activity to respiratory input and, surprisingly, the removal of the OB input results in greater freezing behavior. (This is not so surprising when one considers the results within the olfactory bulbectomy model for testing antidepressants.) My main concern is that the authors have not convinced me that the respiratory signal in the pPFC is not a result of volume conduction from the OB. The two areas are not very far apart, and the very large respiratory signal in the OB might easily be picked up in adjacent cortical areas. See specific comments below.

1. What is the phase difference between the two areas in the respiratory band? The apparent zero (180 deg reversed) phase in Fig 2c is suspicious. Even if differential recordings are used, if the two signals are not completely matched for amplitude and precisely reversed in phase, contaminating volume conduction signals are not completely removed. Phase measurements are important, and zero phase should be treated as volume conduction. Please report phase differences with coherence measures (see #2 below). The best way to rule out volume conduction is by current source density, but if the phase relationship is significantly nonzero (and not 180 degrees difference), or if the rhythm is reversed or partially reversed across the two leads within pPFC, then this is good evidence that the signal is not the result of volume conduction.
2. For Fig. 2d please use coherence measurements instead of cross-correlation. Cross-correlation is subject to biases from large oscillations on one channel. All statistical comparisons should be

done on coherence, not cross-correlation.

3. I am not convinced by Fig 5b,c that OB LFP does not have any respiratory oscillations. Again, coherence is necessary here. If there remains nonzero coherence between LFP and respiration, then this could explain the residual coherence between respiration and pIPFC. The way to determine if coherence values are above zero (chance levels of coherence) is to shuffle the signals (i.e., compute coherence with mismatched segments of data from the same data on which coherence is being measured). I see from the figure that the respiratory signal is probably significantly decreased, but not that it is gone. There may be some residual or regrown sensory input, which would account for this. If a respiratory signal was also recorded in the TTX condition, it would be helpful to see the same analysis with these data (but not necessary).

4. Coherence with respiratory frequency does not mean that OB drives these oscillations in pIPFC. It is possible that the 4Hz respiration rate is an internal matching of a motor program (respiration) to an ongoing oscillatory coupling among motor and limbic areas.

5. I am surprised that the authors do not at all mention the context of olfactory bulbectomy in anxiety-like states, especially in the context of fear conditioning. This is particularly interesting in the TTX experiments. Removal of the OB inputs bilaterally should increase fear/anxiety behavior.

6. One outcome from these data may be an observation that headfixed mice might be in fear state, b/c they breathe at much lower frequencies than freely moving mice (typically around 4 Hz, I think).

Minor

1. The authors claim that the OB respiratory signal may be driven by mechanosensitive OSNs in the nose. This may indeed be correct, but a more parsimonious interpretation is simply that synchronized excitatory input to the glomeruli accompanies sensory nerve input producing a general rise in excitation in the M/T cells in the OB; the glomerular network takes care of it from there, sculpting the signal.

2. Line 123: Fig. c,d needs a figure number

3. Was thermistor on same or opposite side to recordings?

4. Line 623: distributions should be distributions

Reviewer #2 (Remarks to the Author):

In two recent papers, Cyril Herry's group demonstrated a prominent 4-Hz oscillation in prefrontal cortex and BLA during freezing (Dejean et al., 2016, Nature; Karalis et al., 2016, Nature Neuroscience). In the present study, Moberly et al. present data that indicate a strong connection between freezing-related prefrontal oscillatory activity and respiration. This puts the hitherto enigmatic 4-Hz oscillation into new light and reveals much about its origin and potential neural sources. Therefore, the topic is of significant interest. Nevertheless, I have concerns regarding to what extent the data support the authors' conclusions.

Major points

1. My major concern is that the data is not consistently presented and the reader usually has to rely on single examples with no summary data presented. This way it is impossible to judge whether the results were consistent across the data set or not and whether there is enough data to robustly support the claims.

Specifically, in Fig. 1, panel a shows five mice based on the figure; however, the legend claims 6. Panel b is a single example. Where is the corresponding summary data? What proves that respiration was characteristic across multiple freezing bouts of the same mouse as well as across mice? Panel c has 3 mice; what happened to the other 2? or 3? present in panel a? Which mice were included in Fig.2e?

Figure 3 is supposed to show that naris occlusion blocks respiratory rhythms; however, this figure is not convincing in its present form. Panel a is probably a single example – no sample size or measure of variation or error. I could not find corresponding summary data and statistics. Panels d and e show a relatively small effect using a low number of mice. Paired t-test is not exactly appropriate for this comparison, because this type of data cannot be normally distributed. The marginally significant differences would not reach $p=0.05$ with the non-parametric Mann-Whitney U-test. Therefore, a higher sample size is necessary.

Figure 4: what was the sample size for panel c? The SEM looks large – was there any significant effect?

The micrograph in Fig.S1 does not convincingly show fibers in the PL; magnification is very low.

Figure 5: Where is the summary data for decreased respiration-linked oscillation in the OB? Panel b is only example. It might be Fig. S3 – but there is no SD, no sample size and no statistics.

Figure 6 – Based on b, baseline looks as different as tone-evoked freezing. Again, the sample size is too low and the choice of statistics is inappropriate.

Fig.S5 does not seem to have group data, sample size, SD or SEM.

2. Systemic injection of methimazole can have many effects outside the olfactory epithelium, so this experiment is hard to interpret. Since it indirectly contradicts previous results from the Herry lab that 4-Hz oscillation promotes freezing, disruption of olfactory input should be done more specifically to support the claim of increased freezing.

Minor points

1. Fig. 5d – no ‘top’ and ‘bottom’; auto-correlation is mislabelled as cross-correlation
2. What was the reason for using distance as quantification in Fig. 6e compared to freezing time (e.g. Fig.6b)?
3. Relationship to hippocampal oscillations (Fig.S4) is interesting. Was there synchrony (cross-correlations, phase coupling, coherence) between CA1 and PL/OB?
4. Fig.4. “similar results were obtained at other frequencies tested” – please show.
5. Optogenetic stimulation: what was the virus type, amount of virus injected, stimulation pattern used?
6. Which behavior analysis software was used to score freezing?
7. ‘distrubution’ should read distribution (p. 27 legend c)
8. p. 6. Fix figure references.

Reviewer #3 (Remarks to the Author):

The role of breathing-related signals in modulating emotion, memory, and other non-respiratory functions is increasingly being recognized as significant. Here, the authors investigate the effects of perturbing olfactory signals on freezing behavior in mice, which are significant. The experiments are well performed and analyzed. However, the experiments are insufficient to disambiguate whether the effects are due to loss of olfactory signals, modulation of breathing related rhythmicity of these signals, or both. As presented, this paper will be misinterpreted as demonstrating that breathing rhythm (note the term “nasal breathing” used throughout the manuscript, including the title), per se, plays a critical modulatory role in freezing behavior. Either additional experiments will need to be done or the authors will need to revise their

interpretation to be within the bounds of what they observed.

The principal issue is that disruption of olfactory signals via ablation of OSNs or inactivation of the entire olfactory bulb will have broad effects throughout the brain that could result in an increase freezing behavior. While the respiratory-modulated signal is disrupted in the pIPFC, this signal is disrupted everywhere else as well, so the pathway by which respiratory-entrained olfactory signals affect freezing behavior is still unclear. In fact, no data is presented that shows that the respiratory-modulation itself was critical, their data show correlation, not causality. Would the effect still be present in the absence of “nasal breathing”? Can mice be manipulated to breathe orally? If so, the authors might be able better to establish/substantiate their present interpretation.

The title presupposes the authors’ interpretation. A more neutral title may be more appropriate.
-Title: “Nasal Breathing Modulates Rhythmic Activity in the Prefrontal Cortex and Conditioned Fear Behavior“
Rather, “Olfactory sensory signals modulate respiratory-related activity in the prefrontal cortex and conditioned fear behavior”

Introduction

-2nd sentence: “Nasal breathing is commonly effective in suppressing excessive arousal states and fearful emotions and emphasized in yoga and meditative practice4.”

Is it “well known” that nasal breathing in particular is effective in affecting emotion during yoga and meditation, as opposed to oral breathing? The cited reference only discusses controlled breathing in general. Is “nasal breathing” obligatory for the reported effects in humans?

-last sentence: “Remarkably, in the conditioned fear paradigm, mice had prolonged freezing periods when peripheral olfactory signals were removed. These results suggest that respiration-locked olfactory signals have a functional role in the processing of emotional information.”
The olfactory signals have not been shown to have a functional role in processing emotional information. Their data suggest that olfactory sensory information contribute to expression of fear-related behavior.

Results

-Pg 7 “To investigate the consequences of disrupting the olfactory-pIPFC pathway on the fear circuit and behavior, we first used methimazole to lesion the nasal epithelium and thus remove peripheral olfactory inputs for reasons detailed in the Methods”

The authors are not specifically disrupting the olfactory-pIPFC pathway. Rather, they are non-specifically removing the olfactory sensory pathway. Also, the reasons for using methimazole is detailed in the Discussion, not Methods.

-It may be useful to show that respiratory rate remains at 4Hz after methimazole treatment. Also, in Suppl Fig. 3, the baseline pIPFC power distribution looks different at baseline conditions- the control peak at 4Hz is gone, and there is increased power at 6-8Hz in methimazole-treated group. Taken together with the slight increase in % time freezing at baseline, though not significant, this hints at the possibility that there may be some difference in baseline behavior with olfactory epithelium ablation that could explain the increase in freezing. Also, the statistics for retrieval during fear conditioning may require a repeated measures comparison rather than individual t-tests, as the same group is repeatedly sampled.

-Pg. 8 “These findings indicate that disruption of olfactory inputs leads to excessive freezing, suggesting that respiration entrained olfactory signals are involved in the cessation of freezing behavior. “

The results do not show that the olfactory sensory signals play a role in terminating freezing behavior, this should be in the Discussion.

Discussion

-Pg. 10 “However, a few lines of experimental evidence suggest that the 4-Hz oscillation in the pIPFC is not the sole determinate of freezing behavior.“

There was no data in the first place to show that the 4Hz pIPFC oscillation is the sole determinate of freezing behavior.

-“Although the neural signals involved in freezing cessation are not fully understood, termination of the 4-Hz oscillation in the pIPFC is likely involved^{22, 23.}” ... “Changes in the breathing pattern may help to disturb the ongoing 4-Hz oscillation in the pIPFC to terminate freezing.” Unclear why this is the case, or if pIPFC even participates in the increase or termination of freezing behavior.

Pg. 11, end of Discussion “Because the respiration-related rhythm organizes the activity of a variety of brain networks^{14, 15, 16, 44, 46,} it is not surprising that conscious nasal breathing practiced in yoga and meditation is effective in promoting general mental health^{1.} On the other hand, disruption of normal nasal airflow in patients with allergic rhinitis or empty nose syndrome often results in irritability, anxiety, and depression^{47, 48.} Our study suggests a neural basis underlying the mental benefits of controlled nasal breathing by modulating rhythmic activities in critical brain centers.”

It is still unclear if nasal breathing alone (vs regular or oral breathing) is effective in modulating emotions and promoting general mental health. The data is not convincing that changing the rhythmic activities in limbic regions is what provides the benefits of breathing.

Point-by-point Responses (original comments in *italic* and our responses in **Bold**).

Reviewer #1 (Remarks to the Author):

This study addresses respiratory coupling between the OB and prelimbic PFC. The topic is important and timely, given the recent small proliferation of studies in mice and humans showing that there is respiratory coupling between the olfactory system and hippocampus and other cortical areas. The data suggest that respiratory activity modulates freezing behavior. Since recent studies have suggested that 4 Hz stimulation of the plPFC is sufficient to induce freezing, the current report is addressing where that 4 Hz input might come from. They show that mice breathe at 4 Hz when they freeze and that this 4 Hz signal is present in both the OB and plPFC with apparently zero phase. The authors do a nice job of linking this rhythmic activity to respiratory input and, surprisingly, the removal of the OB input results in greater freezing behavior. (This is not so surprising when one considers the results within the olfactory bulbectomy model for testing antidepressants.)

My main concern is that the authors have not convinced me that the respiratory signal in the plPFC is not a result of volume conduction from the OB. The two areas are not very far apart, and the very large respiratory signal in the OB might easily be picked up in adjacent cortical areas. See specific comments below.

1. What is the phase difference between the two areas in the respiratory band? The apparent zero (180 deg reversed) phase in Fig 2c is suspicious. Even if differential recordings are used, if the two signals are not completely matched for amplitude and precisely reversed in phase, contaminating volume conduction signals are not completely removed. Phase measurements are important, and zero phase should be treated as volume conduction. Please report phase

differences with coherence measures (see #2 below). The best way to rule out volume conduction is by current source density, but if the phase relationship is significantly nonzero (and not 180 degrees difference), or if the rhythm is reversed or partially reversed across the two leads within plPFC, then this is good evidence that the signal is not the result of volume conduction.

As suggested, we conducted phase analysis on the OB and plPFC signals which showed a $\sim 13.5^\circ$ difference (equivalent to ~ 10 ms at 4 Hz), significantly different from 0° (new Fig. 2e). The “apparent zero phase” in the original Fig. 2c was due to the long time scale (2 sec) that made the small difference of 10 ms unnoticeable. We also reported phase differences with coherence measures wherever applicable (see #2 below). Although we did not perform systematic current source density analysis, we observed partially reversed rhythm across the two leads (one example is shown below).

2. For Fig. 2d please use coherence measurements instead of cross-correlation. Cross-correlation is subject to biases from large oscillations on one channel. All statistical comparisons should be done on coherence, not cross-correlation.

As suggested, we added coherence measurements in the new Fig. 2c. Since cross-correlation informs different aspects of the similarity of two signals, we keep the cross-correlation analysis in the new Fig. 2d. The statistical comparison for both peak coherence and cross-correlation reaches the same conclusion.

3. I am not convinced by Fig 5b,c that OB LFP does not have any respiratory oscillations. Again, coherence is necessary here. If there remains nonzero coherence between LFP and respiration, then this could explain the residual coherence between respiration and plPFC. The way to determine if coherence values are above zero (chance levels of coherence) is to shuffle the signals (i.e., compute coherence with mismatched segments of data from the same data on which coherence is being measured). I see from the figure that the respiratory signal is probably

significantly decreased, but not that it is gone. There may be some residual or regrown sensory input, which would account for this. If a respiratory signal was also recorded in the TTX condition, it would be helpful to see the same analysis with these data (but not necessary).

We thank the reviewer for this suggestion. We performed the methimazole experiments in additional mice and provided quantitative analysis on peak coherence between the OB LFP and respiration signals (new Fig. 5c). As expected, under control conditions, the coherence between the OB LFP and respiration signals is close to 1.0 between 2-10 Hz (the normal range of breathing rates): 0.92 in the awake state and 0.97 under anesthesia. After methimazole treatment, the peak coherence is significantly reduced (to 0.71 in the awake state and 0.63 under anesthesia). However, the coherence is not completely gone since the average peak coherence between the respiration and shuffled OB signals is 0.55 (chance level coherence). For shuffled OB signals, we tested two approaches: (1) the OB traces were broken up into 250 ms segments then randomly rearranged and concatenated or (2) all data points were completely randomized to form a noise-like signal. Both approaches generated similar peak coherence values at ~0.5. In the new Fig. 5c, the chance level coherence was obtained by using the first approach. Note that the peak coherence between the respiration and OB signals always occurred at the dominant breathing frequency, but for shuffled data, the peak coherence could occur randomly in the full frequency range (between 1-12 Hz).

The residual coherence between the OB LFP and respiration signals could be due to centrifugal respiration-coupled inputs to the OB or potential regrowth of sensory inputs, which would be minimal at this time. This point is included in Results (page 8 last para).

The respiratory signal was not recorded in the TTX condition due to the nearly complete elimination of the OB neural activity after TTX infusion (new Fig. 6d) and the technical challenges of implanting cannula, electrodes and thermocouple in the same animal.

4. Coherence with respiratory frequency does not mean that OB drives these oscillations in plPFC. It is possible that the 4Hz respiration rate is an internal matching of a motor program (respiration) to an ongoing oscillatory coupling among motor and limbic areas.

We agree with the reviewer. In this study, we experimentally tested the contribution of olfactory inputs to the 4-Hz oscillations in the plPFC, but we did acknowledge other potential sources (e.g. intrinsically generated or other respiration-related signals in the brain) in the Results (page 6, last para to page 7, 1st para) and Discussion (page 12, 1st para).

5. *I am surprised that the authors do not at all mention the context of olfactory bulbectomy in anxiety-like states, especially in the context of fear conditioning. This is particularly interesting in the TTX experiments. Removal of the OB inputs bilaterally should increase fear/anxiety behavior.*

We thank the reviewer for pointing this out. Our results are in general agreement with the effects of olfactory bulbectomy on fear/anxiety behavior. We added a brief discussion on this point in both Results (page 10, 1st para) and Discussion (page 13, 1st para).

6. *One outcome from these data may be an observation that headfixed mice might be in fear state, b/c they breathe at much lower frequencies than freely moving mice (typically around 4 Hz, I think).*

This is an interesting point, which would be worth further investigation. Since we did not specifically test head-fixed mice, we prefer not to speculate on whether these animals are in fear state.

Minor

1. *The authors claim that the OB respiratory signal may be driven by mechanosensitive OSNs in the nose. This may indeed be correct, but a more parsimonious interpretation is simply that synchronized excitatory input to the glomeruli accompanies sensory nerve input producing a general rise in excitation in the M/T cells in the OB; the glomerular network takes care of it from there, sculpting the signal.*

The statement “...presumably by activating intrinsically mechanosensitive olfactory sensory neurons (OSNs)...” meant to explain a peripheral source for the synchronized excitatory input to the glomeruli. This does not rule out the contribution of the OB network to the respiratory signals. Two recent papers (Wu et al., 2017, J Neurosci; Iwata et al., 2017, Neuron) confirm the mechanosensitivity of OSNs using in vivo mouse models, which are cited to further support this statement (page 3, 2nd para).

2. *Line 123: Fig. c,d needs a figure number*

Done.

3. *Was thermistor on same or opposite side to recordings?*

Thermocouples were implanted contralateral to OB and pIPFC electrode implantation sites in our unilateral recordings. This is clarified in the methods (page 15, 2nd para).

4. Line 623: distributions should be distributions

Done.

Reviewer #2 (Remarks to the Author):

In two recent papers, Cyril Herry's group demonstrated a prominent 4-Hz oscillation in prefrontal cortex and BLA during freezing (Dejean et al., 2016, Nature; Karalis et al., 2016, Nature Neuroscience). In the present study, Moberly et al. present data that indicate a strong connection between freezing-related prefrontal oscillatory activity and respiration. This puts the hitherto enigmatic 4-Hz oscillation into new light and reveals much about its origin and potential neural sources. Therefore, the topic is of significant interest. Nevertheless, I have concerns regarding to what extent the data support the authors' conclusions.

Major points

1. My major concern is that the data is not consistently presented and the reader usually has to rely on single examples with no summary data presented. This way it is impossible to judge whether the results were consistent across the data set or not and whether there is enough data to robustly support the claims.

We thank the reviewer for this suggestion. In the revision, we added more animals for each experiment and presented summary data wherever applicable. We also consulted with a biostatistician (Mr. Ming Lu from StatConfidence LLC, Philadelphia) on appropriate statistical tests. To determine sample sizes, we followed the "Guidelines for the Care and Use of Mammals in Neuroscience and Behavioral Research"

(<https://www.ncbi.nlm.nih.gov/books/NBK43321/>). For continuous variables (significance level at 0.05 and power at 0.9), we would need a sample size of 6.3 to observe an effect that is twice the standard deviation. We increased the n number to at least 7 for all the behavioral groups. Parametric tests were used when the data passed Shapiro-Wilk normality tests with equal variance between groups. When data failed to pass the normality tests, we used non-parametric tests (Mann-Whitney in the case of unpaired samples and Wilcoxon matched-pairs in the case of paired samples). These were included in Methods (page 20, 1st para).

Specifically, in Fig. 1, panel a shows five mice based on the figure; however, the legend claims 6.

In the new Fig. 1a, we increased the animal number to 8 for the control group and used scatter dots (instead of vertically aligned dots) to avoid overlapping of individual dots (that is why six animals appeared as five dots in the original figure).

Panel b is a single example. Where is the corresponding summary data? What proves that respiration was characteristic across multiple freezing bouts of the same mouse as well as across mice? Panel c has 3 mice; what happened to the other 2? or 3? present in panel a? Which mice were included in Fig.2e?

Panel b is a single example of respiration rate change between baseline and freezing period and the summary data from eight control animals are now presented in the new Fig. 1e, which shows significantly increased 2-6 Hz bandpower during freezing than baseline. In the new Fig. 1c, the instantaneous breathing frequency is shown before and during a representative freezing epoch from all eight animals. The new Fig. 1d summarizes the breathing rates during individual, continuous freezing bouts that are longer than 5 sec (4 to 7 from each animal) from all eight animals. In the new Fig. 2c (the original Fig. 2e), the same eight control mice are included.

Figure 3 is supposed to show that naris occlusion blocks respiratory rhythms; however, this figure is not convincing in its present form. Panel a is probably a single example – no sample size or measure of variation or error. I could not find corresponding summary data and statistics. Panels d and e show a relatively small effect using a low number of mice. Paired t-test is not exactly appropriate for this comparison, because this type of data cannot be normally distributed. The marginally significant differences would not reach $p=0.05$ with the non-parametric Mann-Whitney U-test. Therefore, a higher sample size is necessary.

We performed unilateral naris closure on more animals with a total number of 7. The new Fig. 3a shows the raw data from one animal and the new Fig. 3b shows the power spectra and quantification from all seven animals on open and occluded sides. We think that the paired t-test is appropriate because the open side from each animal served as a within-subject control for the occluded side and the data sets (the peak coherence and cross-correlation values) passed the normality tests with equal variance. Testing the coherence data set ($p = 0.031$) and the cross-correlation data set ($p = 0.016$) using the non-parametric Wilcoxon matched-pairs test yields similar results.

Figure 4: what was the sample size for panel c? The SEM looks large – was there any significant effect?

We performed more optogenetic experiments and showed averaged data of multiple trials from three animals (new Fig. 4d).

The micrograph in Fig.S1 does not convincingly show fibers in the PL; magnification is very low.

We performed new anterograde tracing experiments by unilaterally injecting a cre-dependent ChR2-EYFP virus into the anterior olfactory nucleus (AON) in the Vglut1-Cre mice ($n = 3$) and found projections to the medial prefrontal cortex (both prelimbic and infralimbic areas) on the injected side. The data are presented in the new Fig. 4b. The Results (page 8, 1st para) and Methods sections (page 17, 1st para) are changed accordingly.

Figure 5: Where is the summary data for decreased respiration-linked oscillation in the OB? Panel b is only example. It might be Fig. S3 – but there is no SD, no sample size and no statistics.

In the new Fig. 5 (methimazole treated mice), we included example traces of respiration and the OB LFPs under both awake and anesthetized states (new Fig. 5b). The summary data on peak coherence between the OB and respiration signals for both states are in the new Fig. 5c. The summary data on the coherence, cross-correlation and phase differences between the OB and pIPFC signals are in the new Fig. 5f-h with example traces in the new Fig. 5e. The Results (page 8-9) are revised accordingly.

Figure 6 – Based on b, baseline looks as different as tone-evoked freezing. Again, the sample size is too low and the choice of statistics is inappropriate.

We increased the n numbers in both control ($n = 8$) and methimazole ($n = 9$) treated animals. In addition, we included the unilateral naris closure mice ($n = 7$) in the statistical analysis since they were compared to the same control group. For the baseline, we compared the total freezing time before the onset of the first tone (new Fig. 6a). Because freezing bouts also occur between tones, which are reported as intrinsic fear states (e.g. Karalis et al., 2016), we compared the total freezing time from the onset of the first tone to the end of the trial among the three groups (new Fig. 6b). Since the data sets pass the normality tests, we used one-way ANOVA tests. The results showed no significant differences in the baseline freezing time but significantly increased freezing time during retrieval trials in the methimazole treated mice (new Fig. 6a-c).

Fig.S5 does not seem to have group data, sample size, SD or SEM.

Since the current study focuses on the conditioned fear related freezing behavior, we have decided not to mention the innate fear related freezing behavior and removed Fig. S5. This will be the focus of a future study.

2. Systemic injection of methimazole can have many effects outside the olfactory epithelium, so this experiment is hard to interpret. Since it indirectly contradicts previous results from the Herry lab that 4-Hz oscillation promotes freezing, disruption of olfactory input should be done

more specifically to support the claim of increased freezing.

We initially considered using genetic methods to disrupt olfactory inputs (Discussion on page 13, 2nd para). However, generic knockout of key olfactory signaling molecules typically leads to anxiety and depression-like behaviors, which may alter conditioned fear related freezing. Inducible knockout is likely compromised by the ongoing neurogenesis in the olfactory epithelium. We acknowledge that injection of methimazole might have non-olfactory off-target effects (page 10, 1st para). However methimazole treatment did not change the baseline freezing time (new Fig. 6a) or respiration rates during normal behavior (new Fig. 5d). In addition, we performed the TTX infusion experiments to acutely inactivate the olfactory bulb, which also caused longer freezing periods (new Fig. 6d-h). For more details, see our response to Minor point #2.

Minor points

1. Fig. 5d – no ‘top’ and ‘bottom’; auto-correlation is mislabelled as cross-correlation

The original Fig. 5d was replaced by the new Fig. 5e-g. The labels and legends match the new panels.

2. What was the reason for using distance as quantification in Fig. 6e compared to freezing time (e.g. Fig.6b)?

Because the TTX infusion mice did not have tethered wire during the retrieval test, we were able to use the automatic video analysis software (ANY-maze) to measure the total distance traveled during the baseline (before the onset of the first tone). However we could not do the same for the methimazole treated mice due to the tethered wire – the movement of the wire was often misinterpreted by the software as animal movement. In revision, we added more animals to the saline and TTX treated groups (n = 7 in each) and compared the pre-tone (new Fig. 6e) and post-tone (new Fig. 6f) freezing time as well as the total distance traveled (new Fig. 6h) between these two groups. The Results (page 10, 1st para) and Discussions (page 13, 2nd & 3rd para) are revised accordingly.

3. Relationship to hippocampal oscillations (Fig.S4) is interesting. Was there synchrony (cross-correlations, phase coupling, coherence) between CA1 and PL/OB?

Since [REDACTED] group has extensively investigated the relationship between hippocampal oscillations and respiratory rhythms, we removed this part and the original supplementary Fig. 4 from our study. [REDACTED]

4. Fig.4. “similar results were obtained at other frequencies tested” –– please show.

We removed this statement in the revision. Instead, we focused on one frequency (13 Hz), which is outside the normal respiration range, and showed summary data from 3 animals in the new Fig. 4d.

5. *Optogenetic stimulation: what was the virus type, amount of virus injected, stimulation pattern used?*

This was achieved in genetically modified OMP-ChR2 mice, which were described in Methods (page 14, 2nd para and page 17, last para).

6. *Which behavior analysis software was used to score freezing?*

In most experiments, freezing was scored manually (Methods on page 18, 1st para) due to the tethered wires that prevent accurate, automatic assessment. However, in the TTX treated mice, we compared the results from manual and automatic scoring, which are in very good agreement. We therefore used the freezing time manually scored in all experiments.

7. *‘distrubution’ should rear distribution (p. 27 legend c)*

Done.

8. *p. 6. Fix figure references.*

Done.

Reviewer #3 (Remarks to the Author):

1. The role of breathing-related signals in modulating emotion, memory, and other non-respiratory functions is increasingly being recognized as significant. Here, the authors investigate the effects of perturbing olfactory signals on freezing behavior in mice, which are significant. The experiments are well performed and analyzed. However, the experiments are insufficient to disambiguate whether the effects are due to loss of olfactory signals, modulation of breathing related rhythmicity of these signals, or both. As presented, this paper will be misinterpreted as demonstrating that breathing rhythm (note the term “nasal breathing” used throughout the manuscript, including the title), per se, plays a critical modulatory role in freezing behavior. Either additional experiments will need to be done or the authors will need to revise their interpretation to be within the bounds of what they observed.

The principal issue is that disruption of olfactory signals via ablation of OSNs or inactivation of the entire olfactory bulb will have broad effects throughout the brain that could result in an increase freezing behavior. While the respiratory-modulated signal is disrupted in the plPFC, this signal is disrupted everywhere else as well, so the pathway by which respiratory-entrained

olfactory signals affect freezing behavior is still unclear. In fact, no data is presented that shows that the respiratory-modulation itself was critical, their data show correlation, not causality. Would the effect still be present in the absence of “nasal breathing”? Can mice be manipulated to breathe orally? If so, the authors might be able better to establish/substantiate their present interpretation.

The title presupposes the authors’ interpretation. A more neutral title may be more appropriate.
-Title: “Nasal Breathing Modulates Rhythmic Activity in the Prefrontal Cortex and Conditioned Fear Behavior“

Rather, “Olfactory sensory signals modulate respiratory-related activity in the prefrontal cortex and conditioned fear behavior”

We agree with the reviewer that we can’t disambiguate the effect of “nasal breathing” from that of the broad respiration related rhythm in the brain. Since mice are obligated nasal breathers, it would be difficult (if possible at all) to tease this out in freely behaving animals. We thank the reviewer for the suggestion and changed the title to “Olfactory Sensory Signals Modulate Respiration-Related Rhythmic Activity in the Prefrontal Cortex and Conditioned Fear Behavior”. Throughout the manuscript, we also replaced the phrase “nasal breathing” with “respiration-entrained olfactory signals” (or similar), wherever appropriate.

2. Introduction

-2nd sentence: “Nasal breathing is commonly effective in suppressing excessive arousal states and fearful emotions and emphasized in yoga and meditative practice4.”

Is it “well known” that nasal breathing in particular is effective in affecting emotion during yoga and meditation, as opposed to oral breathing? The cited reference only discusses controlled breathing in general. Is “nasal breathing” obligatory for the reported effects in humans?

Although nasal breathing has been emphasized in these practices and differential neural activities in the human brain are induced by nasal vs oral breathing, we agree that systematic studies to firmly establish the benefits of “nasal” over “oral” breathing are still missing. We therefore rephrased the sentence to “Self-regulation of breathing is effective in suppressing excessive arousal states and fearful emotions and emphasized in yoga and meditative practice.” (page 3, 1st para)

3. -last sentence: “Remarkably, in the conditioned fear paradigm, mice had prolonged freezing periods when peripheral olfactory signals were removed. These results suggest that respiration-locked olfactory signals have a functional role in the processing of emotional information.”

The olfactory signals have not been shown to have a functional role in processing emotional

information. Their data suggest that olfactory sensory information contribute to expression of fear-related behavior.

As suggested, we rephrased this sentence to “These results suggest that respiration-locked olfactory signals contribute to expression of fear-related behavior.” (page 4, last para)

4. Results

-Pg 7 “To investigate the consequences of disrupting the olfactory-plPFC pathway on the fear circuit and behavior, we first used methimazole to lesion the nasal epithelium and thus remove peripheral olfactory inputs for reasons detailed in the Methods”

The authors are not specifically disrupting the olfactory-plPFC pathway. Rather, they are non-specifically removing the olfactory sensory pathway. Also, the reasons for using methimazole is detailed in the Discussion, not Methods.

As suggested, we revised this sentence to “To investigate the consequences of disrupting the olfactory inputs on the fear circuit and behavior, we first used methimazole to lesion the nasal epithelium for reasons detailed in the Discussion”. (page 8, last para)

5. -It may be useful to show that respiratory rate remains at 4Hz after methimazole treatment. Also, in Suppl Fig. 3, the baseline plPFC power distribution looks different at baseline conditions- the control peak at 4Hz is gone, and there is increased power at 6-8Hz in methimazole-treated group. Taken together with the slight increase in % time freezing at baseline, though not significant, this hints at the possibility that there may be some difference in baseline behavior with olfactory epithelium ablation that could explain the increase in freezing. Also, the statistics for retrieval during fear conditioning may require a repeated measures comparison rather than individual t-tests, as the same group is repeatedly sampled.

See our responses to Reviewer #1 Comment 3 and Reviewer #2 Comment 1 on Fig. 5. Now we show example traces and summary data on the OB and respiration coupling before and after methimazole treatment in the new Fig. 5b, c. We also show that methimazole treatment did not change the respiration rates in normal behaviors (new Fig. 5d). The OB-plPFC coherence graph (from nine animals) showed an evident peak at ~4 Hz (Fig. 5f), suggesting that methimazole-treated mice also breathed at ~4 Hz during freezing. Although we did not explicitly show this in all methimazole treated mice (which would require simultaneous thermocouple, OB and plPFC recordings), we did confirm in three methimazole treated mice that they still breathed at ~ 4 Hz during freezing. Compared to control animals, the peak coherence between OB-plPFC signals was significantly reduced (methimazole treated 0.59 ± 0.04 , $n = 9$ vs control 0.88 ± 0.03 , $n = 8$; $U = 1.5$, $p = 0.0002$ in two-tailed Mann-Whitney test; c.f Fig. 5f to Fig. 2c). These data are described in Results (page 8, last para and page 9, 1st para).

In the behavioral tests, we increased the number of methimazole treated mice to nine and the baseline freezing time is not significantly different from the control or unilateral naris closure groups (new Fig. 6a). For statistical analysis, now we compare the total freezing time during retrieval trials since freezing also occurs between tones (Karalis et al., 2016). We used one-way ANOVA and post hoc tests among the three groups (new Fig. 6a, b and page 9, 1st para). For statistical analysis, see also our response to Reviewer 2 Major point #1.

6. -Pg. 8 *“These findings indicate that disruption of olfactory inputs leads to excessive freezing, suggesting that respiration entrained olfactory signals are involved in the cessation of freezing behavior.”*

The results do not show that the olfactory sensory signals play a role in terminating freezing behavior, this should be in the Discussion.

As suggested, we removed this sentence from Results.

7. *Discussion*

-Pg. 10 *“However, a few lines of experimental evidence suggest that the 4-Hz oscillation in the plPFC is not the sole determinate of freezing behavior.”*

There was no data in the first place to show that the 4Hz plPFC oscillation is the sole determinate of freezing behavior.

We removed this sentence and revised the entire paragraph (page 12, 1st para).

8. -*“Although the neural signals involved in freezing cessation are not fully understood, termination of the 4-Hz oscillation in the plPFC is likely involved^{22, 23.}” “Changes in the breathing pattern may help to disturb the ongoing 4-Hz oscillation in the plPFC to terminate freezing.”*

Unclear why this is the case, or if plPFC even participates in the increase or termination of freezing behavior.

Although the necessity of plPFC in expression of the conditioned fear-related freezing behavior has been reported, we agree that these sentences are too speculative. We revised these paragraphs by removing these comments (page 12, last para).

9. Pg. 11, end of Discussion *“Because the respiration-related rhythm organizes the activity of a variety of brain networks^{14, 15, 16, 44, 46,} it is not surprising that conscious nasal breathing practiced in yoga and meditation is effective in promoting general mental health^{1.} On the other*

hand, disruption of normal nasal airflow in patients with allergic rhinitis or empty nose syndrome often results in irritability, anxiety, and depression^{47, 48}. Our study suggests a neural basis underlying the mental benefits of controlled nasal breathing by modulating rhythmic activities in critical brain centers.”

It is still unclear if nasal breathing alone (vs regular or oral breathing) is effective in modulating emotions and promoting general mental health. The data is not convincing that changing the rhythmic activities in limbic regions is what provides the benefits of breathing.

See our response to Question #1. As suggested, this paragraph is revised by tuning down on the benefit of “nasal” breathing (page 13-14).

Reviewers' comments:

Reviewer #1 (Remarks to the Author):

This is a much improved paper after revision. It is clear what the main points are, and the improved statistical methods and numbers make the whole story more solid. The data and results are an important and significant addition to the growing body of literature describing respiratory and olfactory interactions with neocortical circuits. I have only a couple of minor comments.

1. On line 123 and in the abstract, the claim is made that the OB and plPFC LFPs are strongly synchronized at the respiratory frequency. They are not synchronized; there is a robust phase difference between the two (probably due to transmission delay between OB and plPFC). Synchrony implies a zero phase difference and would, in fact, argue for volume conduction. I suggest the authors change the word synchrony to something else, like strongly coherent/coupled/coordinated.

2. The shuffled coherence values in figure 5 allow the reader to understand that there is residual coherence after ablation of the sensory neurons. Can the authors provide a similar shuffled baseline for the coherence as reported in Figure 3d?

3. The claim on the bottom of page 9 (line 200), that "...normal olfactory inputs could reduce freezing time..." could be the opposite as well. Lacking olfactory inputs could increase freezing time.

Reviewer #2 (Remarks to the Author):

The authors have made significant improvements on the manuscript.

1. My first major concern about summary data, statistics and consistency was addressed in a satisfactory way by adding more data and showing summary diagrams, often indicating all data points. As a minor note I would still like to remark that there is no such thing as 'passing the normality test' - normality tests have an alternative hypothesis of violating normality; if the null hypothesis (normality) cannot be rejected that has no statistical meaning in itself.

2. My concern about the off-target effects of methimazole treatment was only partially addressed. As TTX infusion to OB also increased baseline freezing, this suggests that the two manipulations may act on at least partially independent mechanisms. Also, I did not find any test

of the PFC 4 Hz oscillation in the TTX animals - why was this rather obvious question omitted?

Reviewer #3 (Remarks to the Author):

While Moberly et al. have strengthened their datasets and analyses in this revision, and were responsive to reviewer comments regarding instances of overstated language, they have not addressed the fundamental issue that their conclusions are drawn from correlations, and not substantiated by their data.

Moberly et al. show that 4Hz oscillations become prominent in OB and pIPFC during freezing, and that optogenetic stimulation of LFPs in OSNs can entrain OB and pIPFC. Disruptions of the 4Hz LFP signal via nasal epithelium ablation or OB inactivation correlate with increases in freezing duration, but the data do not show that the loss of the pIPFC 4Hz signal causes the longer freezing bouts, and the authors do not present a clear case for how this finding reveals anything about the functional contribution of olfactory inputs to fear behavior. Indeed, “unilateral naris occlusion decreased the coherence and cross-correlation between the OB and the pIPFC signals,” yet “did not affect freezing behavior,” a result that would seem to complicate, if not contradict outright, the authors’ conclusion that olfactory input entrainment of pIPFC is a key part of the behavioral response to fear, i.e., freezing. Instead of grappling with this interesting finding, the authors ignore it, concluding that the “behavioral significance of olfactory inputs to the fear circuit is manifested by prolonged freezing periods when olfactory signals are interrupted.”

The authors seem to be heavily invested in a model in which respiratory-related olfactory signals generate a rhythm to bind activity within the fear circuit to control behavior and interpret their data assuming that their model is correct, i.e., they state on pg 12 line 250 “Olfactory signals modulate rhythmic activities of multiple sites of the fear circuit,” even though this has not been shown. They provide anatomical evidence for how OB signals could do this. However, their experiments do not show whether or how this happens, and there is no direct evidence provided that non-olfactory OB signals are neural substrates of any behavior.

The authors do acknowledge that the 4Hz oscillation in pIPFC may arise from sources other than OB, and that it may be an epiphenomenon during freezing (pg. 12, lines 258-261), but without providing an explanation of why the reader should believe that it isn’t one. Other possibilities exist, too; the effect of OB inactivation on freezing duration could be the result of the loss of odorant cues that contribute to the timing of the fear response. For example, could the absence of a predator odor (regardless of whether there is a modulating breathing rhythm) in and of itself be a cue to reinitiate movement.

No manuscript can address every aspect of a particular question or every possibility raised by experimental results. However, given the lack of strong, empirical evidence for the role posited by the authors, the failure to consider the discrepancies in their results or include experiments designed to distinguish between possible explanations significantly compromises the manuscript.

In light of these comments, the authors need to substantially revise the DISCUSSION to be consistent with their data, carefully delineating its limitations and avoid unwarranted interpretations.

Point-by-Point Responses (the original comments *in italic* and our responses **in Bold**)

All changes are tracked in the word file and page numbers refer to those in a clean copy.

Some changes including shortened title, subheadings in Results, and figure legend of Figure 5 are made to comply with the editorial policies.

Reviewer #1 (Remarks to the Author):

This is a much improved paper after revision. It is clear what the main points are, and the improved statistical methods and numbers make the whole story more solid. The data and results are an important and significant addition to the growing body of literature describing respiratory and olfactory interactions with neocortical circuits. I have only a couple of minor comments.

1. On line 123 and in the abstract, the claim is made that the OB and plPFC LFPs are strongly synchronized at the respiratory frequency. They are not synchronized; there is a robust phase difference between the two (probably due to transmission delay between OB and plPFC). Synchrony implies a zero phase difference and would, in fact, argue for volume conduction. I suggest the authors change the word synchrony to something else, like strongly coherent/coupled/coordinated.

Done as suggested. We did not use the term “synchronization” or alike in the abstract. We would like to mention that many authors use this term to describe two coupled signals with a robust phase difference (such as References #22, 24, 26).

2. The shuffled coherence values in figure 5 allow the reader to understand that there is residual coherence after ablation of the sensory neurons. Can the authors provide a similar shuffled baseline for the coherence as reported in Figure 3d?

We have added the shuffled baseline for the coherence in Figure 3d and changed the figure legend and Method accordingly (Page 19).

3. The claim on the bottom of page 9 (line 200), that “...normal olfactory inputs could reduce freezing time...” could be the opposite as well. Lacking olfactory inputs could increase freezing

time.

As suggested, we have rephrased this sentence to “... lacking olfactory inputs increases freezing time...” (now in Page 10).

Reviewer #2 (Remarks to the Author):

The authors have made significant improvements on the manuscript.

1. My first major concern about summary data, statistics and consistency was addressed in a satisfactory way by adding more data and showing summary diagrams, often indicating all data points. As a minor note I would still like to remark that there is no such thing as ‘passing the normality test’ - normality tests have an alternative hypothesis of violating normality; if the null hypothesis (normality) cannot be rejected that has no statistical meaning in itself.

We agree with the reviewer on this point. We performed the normality test partially to fulfill the requirement in the Nature Communications checklist. We have rewritten this sentence to “... when the data were normally distributed (Shapiro-Wilk normality tests) ...” (Page 19).

2. My concern about the off-target effects of methimazole treatment was only partially addressed. As TTX infusion to OB also increased baseline freezing, this suggests that the two manipulations may act on at least partially independent mechanisms. Also, I did not find any test of the PFC 4 Hz oscillation in the TTX animals - why was this rather obvious question omitted?

In the revised Discussion, we acknowledge that we could not completely rule out the off-target effects of methimazole treatment on mouse behavior and that methimazole and TTX treatments may act on partially independent mechanisms (Page 13).

We did not specifically test the PFC 4 Hz oscillation in the TTX animals for the following reasons. First, we demonstrated the contribution of the olfactory inputs to the PFC 4 Hz oscillation using two independent methods: unilateral naris closure (Figure 3) and methimazole treatment (Figure 5). Second, a previous study (Biskamp et al., 2017; Ref #16) showed that bullectomy eliminated respiration-coupled PFC oscillation under a different behavioral context. Third, it is technically challenging to implant two cannula (TTX infusion to both OBs) plus microwires in both the OB and PFC for LFP recordings. The behavioral tests (TTX and saline control groups shown in Figure 6e-h) were done in mice

with two cannula implanted in the OBs but without microwires. Since these experiments would take considerable efforts but with limited gains (confirming the contribution of the olfactory inputs to the PFC oscillation, which were demonstrated using two other approaches), we therefore did not pursue.

Reviewer #3 (Remarks to the Author):

While Moberly et al. have strengthened their datasets and analyses in this revision, and were responsive to reviewer comments regarding instances of overstated language, they have not addressed the fundamental issue that their conclusions are drawn from correlations, and not substantiated by their data.

Moberly et al. show that 4Hz oscillations become prominent in OB and plPFC during freezing, and that optogenetic stimulation of LFPs in OSNs can entrain OB and plPFC. Disruptions of the 4Hz LFP signal via nasal epithelium ablation or OB inactivation correlate with increases in freezing duration, but the data do not show that the loss of the plPFC 4Hz signal causes the longer freezing bouts, and the authors do not present a clear case for how this finding reveals anything about the functional contribution of olfactory inputs to fear behavior.

We have rewritten the Discussion to highlight that the power of the 4-Hz LFP signal does not positively correlate with the freezing duration and that more experiments are needed to establish any causal effects. We have tuned down on the functional contribution of olfactory inputs to fear circuit and behavior in both Abstract and Discussion (Page 11-12).

Indeed, “unilateral naris occlusion decreased the coherence and cross-correlation between the OB and the plPFC signals,” yet “did not affect freezing behavior,” a result that would seem to complicate, if not contradict outright, the authors’ conclusion that olfactory input entrainment of plPFC is a key part of the behavioral response to fear, i.e., freezing.

We respectfully disagree with this reviewer’s interpretation on the lack of behavioral effects from the unilateral naris occlusion experiments. We do not anticipate that unilateral naris occlusion would affect freezing behavior because the olfactory inputs from the open side are still intact. This was clarified on Page 7.

Instead of grappling with this interesting finding, the authors ignore it, concluding that the “behavioral significance of olfactory inputs to the fear circuit is manifested by prolonged freezing periods when olfactory signals are interrupted.”

The authors seem to be heavily invested in a model in which respiratory-related olfactory signals generate a rhythm to bind activity within the fear circuit to control behavior and interpret their

data assuming that their model is correct, i.e., they state on pg 12 line 250 “Olfactory signals modulate rhythmic activities of multiple sites of the fear circuit,” even though this has not been shown. They provide anatomical evidence for how OB signals could do this. However, their experiments do not show whether or how this happens, and there is no direct evidence provided that non-olfactory OB signals are neural substrates of any behavior.

The authors do acknowledge that the 4Hz oscillation in plPFC may arise from sources other than OB, and that it may be an epiphenomenon during freezing (pg. 12, lines 258-261), but without providing an explanation of why the reader should believe that it isn't one. Other possibilities exist, too; the effect of OB inactivation on freezing duration could be the result of the loss of odorant cues that contribute to the timing of the fear response. For example, could the absence of a predator odor (regardless of whether there is a modulating breathing rhythm) in and of itself be a cue to reinitiate movement.

No manuscript can address every aspect of a particular question or every possibility raised by experimental results. However, given the lack of strong, empirical evidence for the role posited by the authors, the failure to consider the discrepancies in their results or include experiments designed to distinguish between possible explanations significantly compromises the manuscript.

In light of these comments, the authors need to substantially revise the DISCUSSION to be consistent with their data, carefully delineating its limitations and avoid unwarranted interpretations.

We thank the reviewer for raising these points. We have tuned down our claims on the behavioral significance of olfactory inputs to the fear circuit throughout the manuscript and considered alternative mechanisms on how lack of olfactory inputs could influence animal behavior (Page 12-13).

Reviewers' Comments:

Reviewer #2 (Remarks to the Author):

The authors addressed my points to my satisfaction. I have no more concerns.

Reviewer #3 (Remarks to the Author):

The authors have markedly improved this manuscript. That said, I would suggest a few matters that they might consider.

1. The work is an effort to tie in observations about the effects of VOLITIONAL breathing in humans with ordinary breathing (including in response to "fear") in mice. This is a big leap that should be discussed explicitly.

2. Related to 1, there is a big leap between demonstrating that respiratory-modulated olfactory oscillations reach the mIPFC and the conclusion that it is these signals that are largely responsible to the effects of breathing on emotional states. I am not convinced that the presentation is clear in this regard particularly in the organization of the Discussion, which seems to be quite fragmented. This could be a consequence of authors editing the original Discussion rather than rewriting it from scratch. Perhaps, also the authors should explicitly state the hypothesis they are testing.

Re: manuscript NCOMMS-17-19512C

Point-by-Point Responses

REVIEWERS' COMMENTS:

Reviewer #2 (Remarks to the Author):

The authors addressed my points to my satisfaction. I have no more concerns.

Reviewer #3 (Remarks to the Author):

The authors have markedly improved this manuscript. That said, I would suggest a few matters that they might consider.

1. The work is an effort to tie in observations about the effects of VOLITIONAL breathing in humans with ordinary breathing (including in response to "fear") in mice. This is a big leap that should be discussed explicitly.

Response: These two types of breathing are explicitly discussed in the revised version (Page 13).

2. Related to 1, there is a big leap between demonstrating that respiratory-modulated olfactory oscillations reach the mIPFC and the conclusion that it is these signals that are largely responsible to the effects of breathing on emotional states. I am not convinced that the presentation is clear in this regard particularly in the organization of the Discussion, which seems to be quite fragmented. This could be a consequence of authors editing the original Discussion rather than rewriting it from scratch. Perhaps, also the authors should explicitly state the hypothesis they are testing.

Response: We thank the reviewer for raising these interesting points. The Discussion is rewritten to reduce fragmentation (Page 10-13) and incorporate these points in the last paragraph (Page 13).